# Hierarchical graph learning for protein–protein interaction

Ziqi Gao [1,2], Chenran Jiang[3], Jiawen Zhang[1], Xiaosen Jiang[4], Lanqing Li [5], Peilin Zhao[5], Huanming Yang [4], Yong Huang [6] ✉ & Jia Li [1,2] ✉

Protein-Protein Interactions (PPIs) are fundamental means of functions and signalings in biological systems. The massive growth in demand and cost associated with experimental PPI studies calls for computational tools for automated prediction and understanding of PPIs. Despite recent progress, in silico methods remain inadequate in modeling the natural PPI hierarchy. Here we present a double-viewed hierarchical graph learning model, HIGH-PPI, to predict PPIs and extrapolate the molecular details involved. In this model, we create a hierarchical graph, in which a node in the PPI network (top outside-of-protein view) is a protein graph (bottom inside-of-protein view). In the bottom view, a group of chemically relevant descriptors, instead of the protein sequences, are used to better capture the structure-function relationship of the protein. HIGH-PPI examines both outside-of-protein and inside-of-protein of the human interactome to establish a robust machine understanding of PPIs. This model demonstrates high accuracy and robustness in predicting PPIs. Moreover, HIGH-PPI can interpret the modes of action of PPIs by identifying important binding and catalytic sites precisely. Overall, "HIGH-PPI [https://github.com/zqgao22/HIGH-PPI]" is a domain-knowledge-driven and interpretable framework for PPI prediction studies.

Biological functions are accomplished by interactions and chemical reactions among biomolecules. Among them, protein–protein interactions (PPIs) are arguably one of the most important molecular events in the human body and are an important source of therapeutic interventions against diseases. A comprehensive dictionary of PPIs can help connect the dots in complicated biological pathways and expedite the development of therapeutic[1,2]. In biology, hierarchy information has been widely exploited to gain in-depth information about phenotypes of interest, for example, in disease biology[3–5], proteomics[6–8], and neurobiology[9–11]. Naturally, PPIs encapsulate a two-view hierarchy: on the top view, proteins interact with each other; on the bottom view, key amino acids or residues assemble to form important local domains. Following this logic, biologists often take hierarchical

approaches to understand PPIs[12,13]. Experimentally, scientists often employ high-throughput mapping[14–16] to pre-build the PPI network at scale, and use bioinformatics clustering methods to identify functional modules of the network (top view). On the individual protein level, isolation methods, such as co-immunoprecipitation[17], pull-down[18], and crosslinking[19] are used to establish the structures of individual proteins, so that surficial 'hotspots' can be located and analyzed. In short, hierarchy knowledge of structure information is important to understand the molecular details of PPIs.

More recently, the massive growth in the demand and the cost of experimentally validating PPIs make it impossible to characterize most unknown PPIs in wet laboratories. To map out the human interactome efficiently and inexpensively, computational methods are increasingly

[1]Data Science and Analytics, The Hong Kong University of Science and Technology, Guangzhou 511400, China. [2]Division of Emerging Interdisciplinary Areas, The Hong Kong University of Science and Technology, Hong Kong SAR, China. [3]Pingshan Translational Medicine Center, Shenzhen Bay Laboratory, Shenzhen 518118, China. [4]The Cancer Hospital of the University of Chinese Academy of Sciences (Zhejiang Cancer Hospital), Chinese Academy of Sciences, Hangzhou 310022, China. [5]AI Lab, Tencent, Shenzhen 518000, China. [6]Department of Chemistry, The Hong Kong University of Science and Technology, Hong Kong SAR, China. ✉e-mail: yonghuang@ust.hk; jialee@ust.hk

being used to predict PPIs automatically. Over the past decade, as one of the most revolutionary tools in computation, Deep Learning (DL) methods, have been applied to study PPIs. Development in this field has been mostly focused on two aspects, learning appropriate protein representations[20,21] and inferring potential PPIs by link predictions[22,23]. The former focuses on extracting structural information using protein sequences. In particular, Convolutional Neural Networks (CNNs)[24,25] and Recurrent Neural Networks (RNNs)[26–28] have demonstrated high generalization and fast inference speed to capture key sequence fragments for PPIs[29]. 3D CNNs[21,30,31] have shown to be better at extracting 3D structural features of proteins and thus capturing the spatial-biological arrangements of residues[32] that are important to PPI predictions. However, 3D CNN suffers from high computational burdens and limited resolution that is prone to quantization errors[29]. The latter aspect of DL in PPI predictions focuses on the PPI network structures, which involves developing link prediction methods to identify missing interactions within the known network topology. Link prediction methods based on common neighbor (CN)[33] assign high probabilities of PPI to protein pairs that are known to share common PPI partners. CN can be generalized to consider neighbors from a greater path length (L3)[22], which captures the structural and evolutionary forces that govern biological networks such as the interactome. Additionally, distance-based methods measure the possible distances between protein pairs, such as Euclidean commute time (ECT)[34] and random walk with restart (RWR)[35]. Most methods of traditional link prediction focus on known interactions but tend to overlook important network properties such as node degrees and community partitions.

More importantly, these methods perceive only one of the two views of outside-of-protein and inside-of-protein. Few can model the natural PPI hierarchy by connecting both views. To address this issue, we present a hierarchical graph that applies two Graph Neural Networks (GNNs)[36,37] to represent protein and network structures, respectively. In this way, the limitations of 3D CNN and link prediction methods mentioned above can be circumvented. First, GNNs can learn the protein 3D structures on more efficient graph representations, even when facing high-resolution requirements for structure processing. Second, due to the propagation mechanism, GNNs are capable of recovering network properties such as node degrees and community partitions. In short, this hierarchical graph approach aims at modeling the natural PPI hierarchy with more effective and efficient structure perceptions.

Here we describe a generic DL platform tailored for predicting PPIs, Hierarchical Graph Neural Networks for Protein–Protein Interactions (HIGH-PPI). HIGH-PPI models the structural protein representations with the bottom inside-of-protein view GNNs (BGNN) and the PPI network with the top outside-of-protein view GNNs (TGNN). In the bottom view, HIGH-PPI constructs protein graphs by treating amino acid residues as nodes and physical adjacencies as edges. Thus, BGNN integrates the information of protein 3D structures and residue-level properties in a synergistic fashion. In the top view, HIGH-PPI constructs the PPI graph by taking protein graphs (the bottom view) as nodes and interactions as edges and learns protein–protein relationships with TGNN. In an end-to-end training paradigm, HIGH-PPI gains mutual benefits from both views. On the one hand, the bottom view feeds protein representations to the top view to learn accurate protein relationships. On the other hand, protein relationships learned by the top view provide insights to further optimize the bottom view to establish better protein representations. HIGH-PPI outputs the probabilities of interactions for given protein pairs and predicts key "contact" sites for such interactions by calculating residue importance. We show the effectiveness of HIGH-PPI on the human interactome from the STRING database[38] and compare it with leading DL methods. We demonstrate the superiority of HIGH-PPI with higher prediction accuracy and better interpretability. We also show examples that

HIGH-PPI can identify binding and catalytic sites with high precision automatically.

## Results

### HIGH-PPI introduces a hierarchical graph for learning structures of proteins and the PPI network

Although deep learning (DL) models for Protein–Protein Interaction (PPI) prediction have been studied extensively, it has not yet been developed for simulating the natural PPI hierarchy. Here, we suggest HIGH-PPI, a hierarchical graph neural network, for accurate and interpretable PPI prediction. HIGH-PPI works like biologists in a hierarchical manner as it contains the bottom inside-of-protein view and top outside-of-protein view (schematic view in Fig. 1c and detailed architecture in Supplementary Fig. 1a). On one hand, HIGH-PPI applies the bottom view when dealing with a protein, where a protein is represented by a protein graph with residue as nodes and their physical adjacencies as edges. On the other hand, from the top view, protein graphs and their interactions are considered nodes and edges of the PPI graph, respectively. Correspondingly, two GNNs are respectively employed to learn from protein graphs in the bottom view (BGNN) and learn from a PPI graph in the top view (TGNN). Consequently, a set of graphs are interconnected by edges in a hierarchical graph, to present a potent data representation.

In the proposed end-to-end model, the initial stage is to create protein graphs for learning appropriate protein representation. An adjacency matrix of a protein graph is derived from a contact map connecting physically close residues (See Section 4.1 in "Methods" for details). Node attributes are defined with residue-level features for expressing the physicochemical properties of proteins (See Section 4.1 in "Methods" for details). To produce a protein graph representation, Graph Convolutional Network (GCN)[36] is used in BGNN to optimize the protein graphs. As shown in Fig. 1c, BGNN contains two GCN blocks, and we construct three components for each GCN block to obtain a fixed-length embedding vector for a protein graph. Both the adjacency matrix and the residue-level features matrix are inputs for a GCN layer. To respectively improve model expressiveness and accelerate training convergence, the nonlinear activation function of ReLU and Batch Normalization (BN) are used. Readout operation including a self-attention graph (SAG) pooling[39] and the average aggregation is used to ensure a fixed-length embedding vector output. Regardless of the number and permutation of residues, a 1D embedding vector is obtained after two GCN blocks. By the end of those operations, the final protein representations are assembled, which are employed as initial features of the PPI graph. In TGNN, features are propagated along interactions in the PPI network for learning network community and degree properties. In the top view, we specifically design a GIN block that contains a Graph Isomorphism Network (GIN)[37] layer, ReLU activation function and a BN layer. Node features of the PPI graph are updated with recursive neighborhood aggregations of three GIN blocks. Two arbitrary protein embeddings are combined by concatenation operations, and a Multi-Layer Perceptron (MLP) is then applied as a classifier for prediction. Moreover, we also consider graph attention (GAT) and arbitrarily deploy two of the three GNN layers (i.e., GCN, GIN and GAT) on BGNN and TGNN. The performance of HIGH-PPI with various GNN layers is shown in Supplementary Fig. 2.

We train and evaluate HIGH-PPI on multi-type human PPIs from the STRING database[38], which contains a critical assessment and integration of PPIs. SHS27k[26], a homo sapiens subset from STRING[38] that comprises 1,690 proteins and 7,624 PPIs, is used to train and evaluate the HIGH-PPI unless otherwise noted. However, a small fraction of proteins (∼ 8%) sometimes need to be removed because of the lack of their native structures in the PDB database. While evaluating the prediction performance for multi-type PPIs, we consider the prediction for each PPI type as a one-vs-all binary classification problem, for which two metrics, F1 score and area under the precision-recall curve (AUPR)

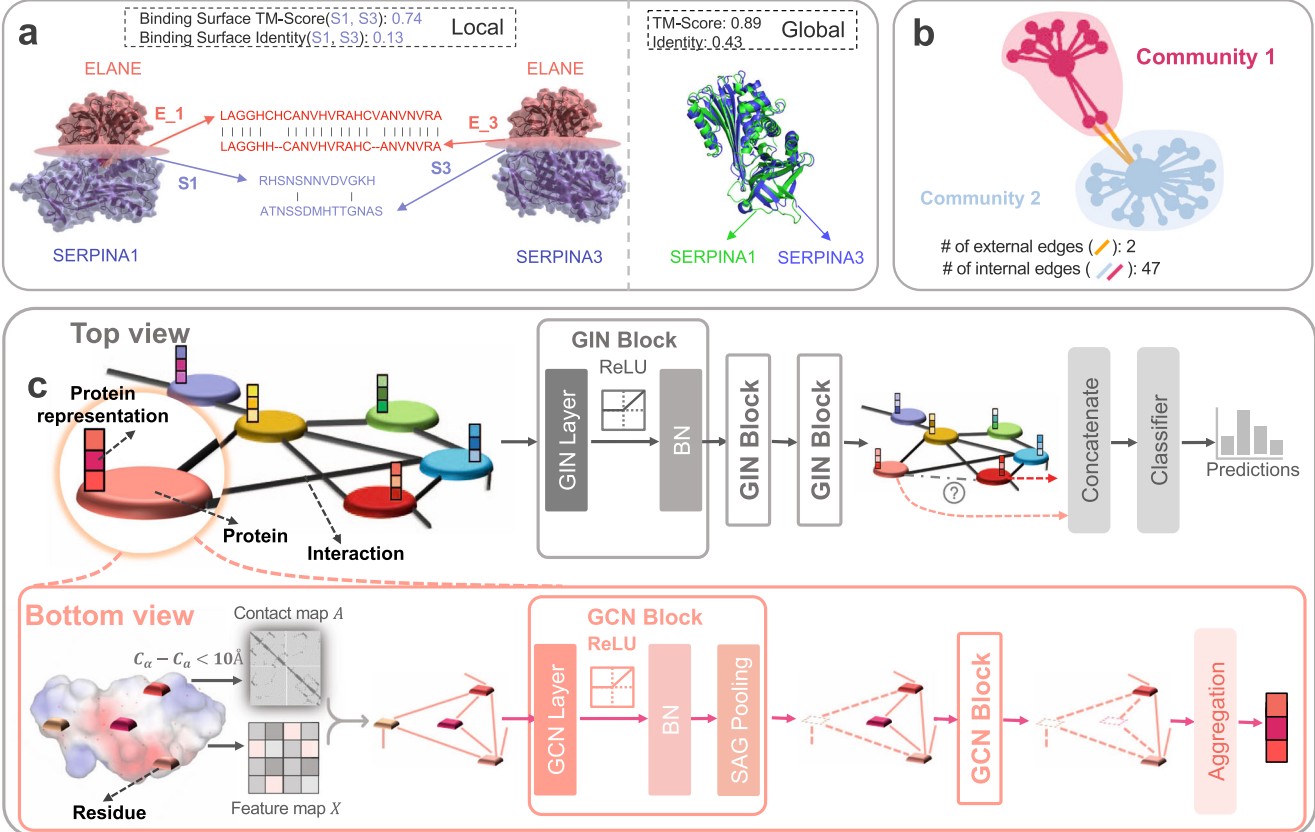

**Fig. 1 | Schematic view of the HIGH-PPI architecture.** Both the protein structure (biology structure) and network structure (interactome structure) are essential for predictions of PPIs. **a** The PPIs with protein structure information. Although protein sequence usually provides details among PPIs, it can also lead to low predictability for PPI prediction. Left: As an example, SERPINA1 and SERPINA3, protein members of a shared superfamily, bind to almost the same binding surface (TM-score is 0.74) of ELANE, whereas they share low sequence consistency (identity is 0.13) locally in the binding surface. Right: From a global perspective, gaps in the sequence and structure of proteins also exist. SERPINA1 and SERPINA3 highly align in structure (TM-Score is 0.89), but share a low sequence consistency (identity is 0.43). **b** The PPIs with network structure information. PPI networks tend to yield community structures that divide proteins into groups with dense connections internally (internal edges) and sparse connections externally (external edges). **c** The HIGH-PPI is a hierarchical model for learning both protein structure information and network structure information. The HIGH-PPI contains two views, the top view and the bottom view. In bottom view, residues serve as nodes, residue-level physico-chemical properties as node features and edges connect physically adjacent residues. Two trainable graph convolutional blocks are applied for learning complex protein representations. In top view, proteins serve as nodes, interactions as edges and representations from the bottom view as node features. Three trainable graph isomorphism blocks are applied to update protein representations and after concatenating a pair of query proteins, the resulting embedding is passed through the linear classifier to learn protein correlations.

are used for predicting the presence or absence of the corresponding PPI class. The overall performance of micro-F1 and AUPR scores for multi-type PPI prediction is averaged across all PPI types.

## HIGH-PPI shows the best performance, robustness and generalization

To validate the predictive power of our model, we compare HIGH-PPI with leading methods from four perspectives, including (1) the overall performance under a random data split, (2) the robustness of HIGH-PPI against random interaction perturbation, (3) model generalization for predicting PPI pairs containing unknown proteins, (4) evaluations in terms of AUPR on five separate PPI types. For each method, all the proposed modules and strategies are involved to get the best performance.

First, we compare the overall performance of HIGH-PPI with leading baselines in Fig. 2a. To ensure native PDB structures for all proteins, we filter from SHS27k and construct the dataset containing ~1600 proteins (see Supplementary Data File 1) and ~6600 PPIs. We randomly select 20% PPIs for testing and compare PPI to one state-of-the-art DL method (i.e., GNN-PPI[24]), one sequence-based method (i.e., PIPR[26]), one 2D CNN-based method (i.e., DrugVQA[40]) and one machine learning (ML) method based on random forest (i.e., RF-PPI[41]). GNN-PPI

applies a GNN module to learn the PPI network topology and 1D CNN to learn protein representations by taking pre-trained residue embeddings as inputs. PIPR, an end-to-end framework based on recurrent neural networks (RNN), represents proteins with only pre-trained residue embeddings. DrugVQA applies a visual question-answering mode to learn from protein contact maps with a 2D CNN model and extract semantic features with a sequential model. Supplementary Data File 2 contains predictions of HIGH-PPI for all test PPIs from SHS27k. We provide the precision-recall curves in Fig. 2a. In terms of best micro-F1 scores (best-F1), HIGH-PPI obtains the best performance. Pre-trained residue embedding method GNN-PPI takes the second place by effectively generalizing to unknown proteins. Without using any pre-training techniques, HIGH-PPI surpasses GNN-PPI by an average of ~4%, showing the superiority of the hierarchical modeling approach. DrugVQA gets relatively poor performance (best-F1 ≈ 0.7), which could be attributed to the neglect of residue property information and structures of the PPI network.

Second, to evaluate the robustness of HIGH-PPI, we analyze the model tolerance against interaction data perturbation including random addition or removal of known interactions. This simulates scenarios where PPI datasets always omit undiscovered interactions and may introduce mislabeled ones. Based on the perturbated PPI network,

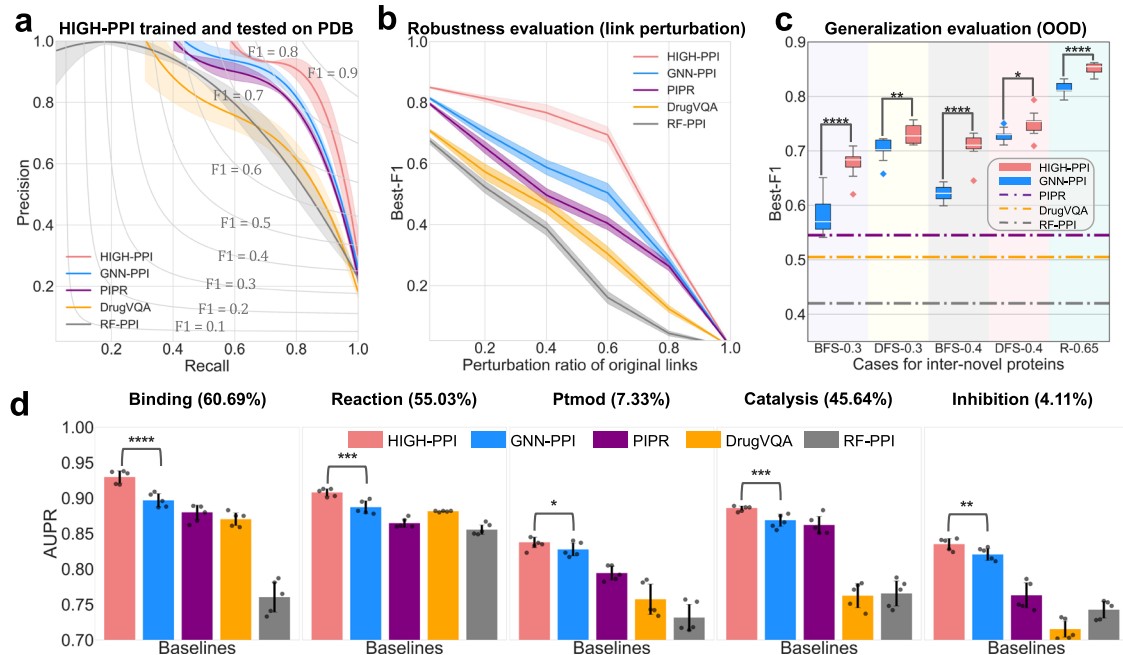

**Fig. 2 | Performance of HIGH-PPI in predicting PPIs. a** Precision-recall curves of PPI prediction on SHS27k (sub-dataset from STRING) containing ~6600 PPIs and ~1500 human proteins with native PDB structures showing the performance of HIGH-PPI compared to baselines containing GNN-PPI, PIPR, DrugVQA and RF-PPI. **b** Robustness evaluation showing the best micro-F1 scores (Best-F1) of baseline predictions against link perturbations of various cases where links are randomly added or removed with different ratios. Error bands of **a** and **b** represent the standard deviation of the mean under 9 independent runs. **c** Generalization evaluation showing Best-F1s of baselines tested on a regular and 4 Out-of-Distribution (OOD) cases, in which datasets are constructed with random split (R), Breath-First Search (BFS) and Depth-First Search (DFS) and three ratios represent probabilities of overlap of proteins between the training and test datasets. Distributions of Best-F1s under 9 independent runs of HIGH-PPI and the second-best baseline (GNN-PPI)

are represented as boxplots (center line, the median; upper and lower edges, the interquartile range; whiskers, $0.5\times$ interquartile range) and moreover, dotted lines show the mean results of 9 independent runs of PIPR, DrugVQA and RF-PPI under DFS-0.4, the easiest OOD pattern. The significance of HIGH-PPI versus GNN-PPI is shown in each case (Two-sided $t$-test results: ****$P = 1.1\times10^{-5}$ for BFS-0.3, ***$P = 4.5\times10^{-3}$ for DFS-0.3, *$P = 1.0\times10^{-7}$ for BFS-0.4, ***$P = 2.0\times10^{-2}$ for DFS-0.4 and **$P = 3.0\times10^{-6}$ for R-0.65). **d** Distributions of AUPR scores of 5 independent runs computed on 5 PPI types and corresponding proportions. Each figure shows the performance significance of HIGH-PPI versus the second-best baseline (GNN-PPI) (Two-sided $t$-test results: ****$P = 2.0\times10^{-5}$ for binding, ***$P = 1.7\times10^{-4}$ for reaction, *$P = 4.4\times10^{-2}$ for ptmod, ***$P = 3.2\times10^{-4}$ for catalysis and **$P = 6.0\times10^{-3}$ for inhibition). Error bars represent standard deviation of the mean. Source data are provided as a Source Data file.

we split the training and test sets at an 8:2 ratio. We observe in Fig. 2b that our method exhibits stable performance in terms of best-F1 with a random perturbation of 40%. When compared to the second-best baseline (i.e., GNN-PPI), HIGH-PPI offers a significant performance gain of up to 19%, which demonstrates the strongest model robustness among all methods. It is crucial to notice that although RF-PPI and DrugVQA perform consistently in the overall evaluation (see Fig. 2a), DrugVQA performs significantly more robustly than RF-PPI, demonstrating the undisputed superiority of DL methods over ML ones. Furthermore, we perform false discovery on our method, which investigates the effect of the training data unreliability (i.e., false negative (FN) and false positive (FP)) on our model and a solid baseline (GNN-PPI). Specifically, we consider the original dataset to be reliable and artificially add perturbations to represent data unreliability. Supplementary Table 1 shows the created 9 datasets with different FP rates ($FPR_{train}$) and FN rates ($FNR_{train}$). We respectively train the model on the reliable training set and created 9 unreliable ones and present the FP rates ($FPR_{pre}$), FN rates ($FNR_{pre}$) and false discovery rates ($FDR_{pre}$) metrics on the test sets (see Supplementary Table 2 and 3). Without unreliability, our model achieves best performance with insignificant superiority (*$P = 3.8\times10^{-2}$) in the $FPR_{pre}$ metric, and considerable superiority in the $FNR_{pre}$ (***$P = 1.2\times10^{-4}$) and $FDR_{pre}$ (***$P = 1.5\times10^{-4}$) metrics. When introducing data unreliability, we are surprised to find that our model substantially improves the superiority significance in the $FPR_{pre}$ metric (****$P = 4.0\times10^{-5}$) while retaining the original significance in $FNR_{pre}$ and $FDR_{pre}$. In addition to showing the excellent robustness of our model, we also provide more in-depth insights in Section 3.2.

Generalization ability is investigated by testing HIGH-PPI in various out-of-distribution (OOD) scenarios where unknown proteins arrive in the test sets with different probabilities (see Fig. 2c). For example, BFS-0.3 denotes that the test set involves 30% known proteins via Breath-First Search approach[24]. For PIPR, DrugVQA and RF-PPI, we visualize their best performances among all OOD cases using dotted lines, to demonstrate the absolute dominance of HIGH-PPI and GNN-PPI. Furthermore, we observe that HIGH-PPI consistently outperforms GNN-PPI, the second-best method, with large margins in all five scenarios. BFS typically produces worse performance than DFS, because BFS creates a more challenging and realistic mode where unknown proteins exist in cluster forms. ML method (RF-PPI) exhibits poor generalization. Furthermore, we follow Park and Marcotte[42] to explore the differences in model performance on 3 kinds of PPI pairs with different degrees of OOD. Specifically, $C_1$ stands for the percentage of PPIs of which both proteins were present in a training set (Class 1), $C_2$ stands for the percentage of PPIs of which one of (but not both) proteins was present in the training set (Class 2), $C_3$ stands for the percentage of PPIs of which neither protein was present in the training set (Class 3). The detailed experimental protocol has been presented in the Supplementary Method 3. We come to the same conclusion as Park and Marcotte did[42]. There is a noticeable difference in model test performance across the 3 distinct classes of test pairs. Particularly, on Class 1 test pairs, both models (HIGH-PPI and GNN-PPI) perform the best, on Class 2 test pairs they are the second best, and on Class 3 test pairs they are the poorest. Furthermore, we find that for each model, the class proportion (i.e., $C_1/C_2/C_3$) had an impact on the overall performance of the model despite having little effect on performance

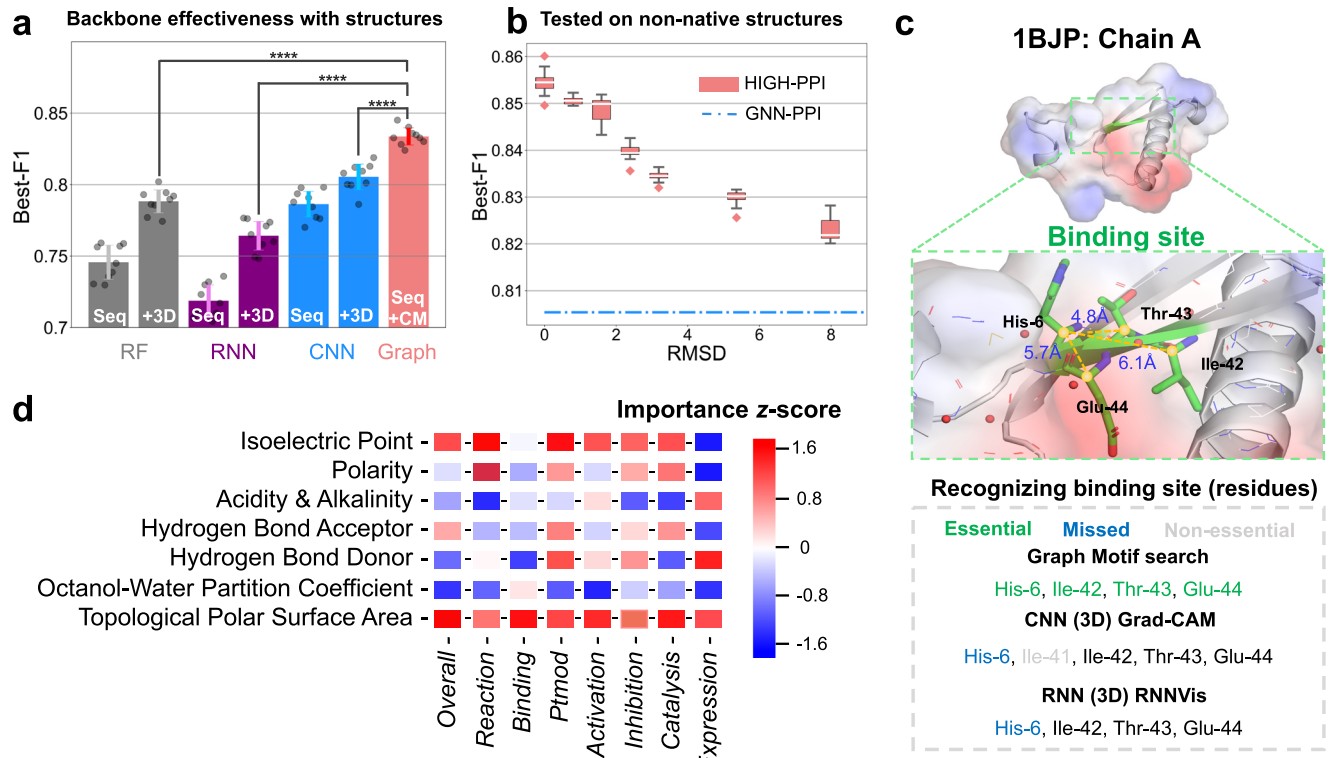

**Fig. 3 | Performance of bottom view GNN of HIGH-PPI to represent a protein for PPI prediction. a** Effectiveness in demonstration w or w/o protein 3D information (3D coordinates of $C_\alpha$ atoms in all residues). The protein is represented with backbones including Random Forest (RF from RF-PPI, gray), Recurrent Neural Networks (RNN from PIPR, purple), Convolutional Neural Networks (CNN (Seq) from GNN-PPI, CNN ( + 3D) from DeepRank blue), respectively. Converting 3D information into protein contact maps (CM), a backbone with graph structured data outperforms all other methods with high performance significances (Two-sided *t*-test results: graph versus RF ( + 3D) ****$P = 1.1 \times 10^{-8}$, graph versus RNN ( + 3D) ****$P = 6.1 \times 10^{-12}$, graph versus CNN ( + 3D) ****$P = 2.3 \times 10^{-7}$). Error bars represent standard deviation of the mean under 9 independent runs. **b** HIGH-PPI can outperform other baselines without absolutely precise structures of query proteins. Blue dotted line (mean value of 9 independent runs) representing the Best-F1 score of second-best baseline (GNN-PPI) without 3D information and boxplot (9 runs with independent seeds) showing the relationship between Best-F1 scores of HIGH-PPI and the Root-Mean-Square Deviation (RMSD) of the tested structures relative to the native structures. For boxplots, the center line represents the median, upper and lower edges represent the interquartile range, and the whiskers represent 0.5× interquartile range. As an example, **c** HIGH-PPI can easily identify the binding site containing four physically adjacent residues via conventional graph motif research method (PDB id: 1BJP). CNN and RNN based backbones may miss (missed) or mis-identify (non-essential) residues with Grad-CAM and RNNVis. **d** The feature importance in residue-level for overall (left-most column) and type-specific (right six columns) PPI prediction calculated as the average *z*-score resulting from dropping each individual feature dimension from our model and calculating changes of AUPR before and after. Source data are provided as a Source Data file.

on the respective classes. Thus, it seems that the proportion of the three test pair classes (Supplementary Table 6) as well as the percentage of unknown proteins (Fig. 2c) in the test sets may both have a significant role in determining the degree of OOD in the dataset.

Finally, for each of the five PPI types, we offer a separate performance analysis in terms of AUPR. In all five types, HIGH-PPI consistently beats other baselines with high significance as shown in Fig. 2d. As anticipated, PPI types with high proportions (such as binding, reaction, and catalysis) can be predicted more easily since the model could learn enough relevant information. In addition, we find that when predicting binding PPIs, HIGH-PPI outperforms GNN-PPI most significantly (****$P = 2.0 \times 10^{-5}$). This is reasonable as HIHG-PPI is designed to recognize spatial-biological patterns of proteins, which is highly related to binding type PPIs. Similar trends are also found in the performance of HIGH-PPI and GNN-PPI in various PPI types under OOD cases (Supplementary Fig. 5).

**Bottom inside-of-protein view improves the performance**
We investigate the role of the bottom inside-of-protein view from four perspectives, including (1) the effectiveness of graph representations and backbones with native protein structures, (2) the model tolerance with low-quality protein structures, (3) the capability to predict motifs (i.e., functional sites) in a protein, (4) the overall and type-specific feature importance.

First, we explore the effectiveness of backbones including RF, RNN, CNN and GNN in Fig. 3a. For fairness, we feed the same features of residue sequence to RF, RNN and CNN, whose results are displayed by bar charts with 'Seq'. We directly use RF-PPI as the RF backbone. For RNN and CNN backbones, we respectively employ the RNN module of PIPR and the CNN module of GNN-PPI to extract sequence embeddings for representing proteins and apply the same fully connected layer as classifiers. We test the predictive power of each model with 3D information. For RF and RNN, we employ the concatenations of sequence data and Cartesian 3D coordinates of each $C_\alpha$. For CNN, we apply the 3D CNN module suggested in DeepRank[21], a deep learning framework for identifying interfaces of PPIs. For GNN, we learn from protein graphs in which the adjacency matrix is determined by $C_\alpha - C_\alpha$ contact map. With the aid of 3D information, we discover all the model performance can be improved, indicating that 3D information is an important complement to sequence-alone information. Importantly, GNN performs the best when compared to RF ( + 3D), RNN ( + 3D) and CNN ( + 3D), which shows that GNN is the best approach for capturing spatial-biological arrangements of residues within a protein. Moreover, GNN performs significantly better than 3D CNN in memory and time efficiency (Supplementary Fig. 3).

Second, we examine the model tolerance when testing with low-quality structure data (see Fig. 3b). This meets the realistic scenarios, where native structure information is not always available for

predicting PPIs. We prefer the model whose performance is not seriously limited by the structure quality, which is robust to inputs directly from computational models (e.g., AlphaFold[43]). We evaluate the quality of the input protein structure by calculating the root-mean-square deviation (RMSD) of the native one and the input. Native protein structures (RMSD = 0) are retrieved from the PDB database at the highest resolutions. We compute the best-F1 scores (box plots) of our method on a set of AlphaFold structures with various RMSDs (0.80, 1.59, 2.39, 3.19, 5.36, 7.98), and show the average result of second-best method (GNN-PPI) in a blue dotted line. As can be seen, our model performance is always better than GNN-PPI, even with RMSD up to 8. The comparison with 3D CNN model[21] further proves the denoising ability of the hierarchical graph for protein structure errors (Supplementary Fig. 4a). In short, our model performance is not significantly affected by structure errors where powerful pre-trained features are not available.

Further, to interpret decisions made by RNN, CNN and GNN, an experiment is conducted to explore the ability to capture protein functional sites. We apply the 3D-grad-CAM approach[44] on the trained 3D CNN model named DeepRank[21], and apply the RNNVis approach[45] on the trained PIPR[26] model with 3D information. All three methods have identified more than one motif, in which we only show the most crucial site. Figure 3c displays the binding site for an isomerase protein's chain A (PDB id: 1BJP). The binding site is made up of four residues with the sequence numbers 6, 42, 43, and 44. As can be seen, whereas neither CNN nor RNN can identify the His-6 residue, our method can precisely identify the binding site by using graph motif search. It seems to be a challenge for the sequence model (i.e., RNN, CNN) to connect His-6 to the other residues, probably because of their weak connections in a sequence mode. Moreover, 3D CNN performs even worse than RNN as it incorrectly classifies the non-essential Ile-41 residue.

For node features in protein graphs, we select seven important features from twelve residue-level feature options (see Supplementary Table 4) that are easily available. The feature selection process (see Supplementary Method 1 for details) produces the optimal set consisting of seven features to ensure that our model peaks at both AUPR and best-F1 scores. Here, we list the selected seven residue-level physicochemical properties in Fig. 3d and discuss their importance for different types of PPIs to both better interpret our model and discover enlightening biomarkers for PPI interface. The average z-score, which results from deleting each feature dimension and analyzing changes in AUPR before and after, is calculated to determine the importance of a feature. We choose a representative type (i.e., binding) to explain because it is the most prevalent in the STRING database. As a consequence, HIGH-PPI regards topological polar surface area (TPSA) and octanol-water partition coefficient (KOW) as dominant features. This finding supports the conventional wisdom that TPSA and KOW play a key role in drug transport process[46], protein interface recognition[47,48], and PPI prediction[49].

## Top outside-of-protein view improves the performance
We investigate the role of top outside-of-protein view TGNN from three perspectives, including (1) the importance of degree and community recovery for predicting network structures, (2) comparison results of TGNN and other leading link prediction methods, (3) a real-life example to show the shortcomings of the leading link prediction methods.

Recently, various works have demonstrated the usefulness of structure properties (e.g., degree, community) of networks for predicting missing links. HIGH-PPI is inspired to efficiently recover the degree and community partitions of the PPI network by utilizing the network topology. We show an empirical study in Fig. 4a to illustrate the impact of degree and community recovery for link prediction. We randomly select the test results from the model trained in different

epochs and calculate the negative Mean Absolute Error (-MAE) of the predicted degrees and real degrees to represent degree recovery. Similarly, for community recovery, we quantify the community recovery using the normalized mutual information (NMI). As can be seen, we observe a significant correlation ($R = -0.66$) between degree recovery and model performance (i.e., best-F1) as well as a high correlation ($R = 0.68$) between community recovery and model performance, which means better recovery of the degree and community of PPI network implies better PPI prediction performance.

Second, we evaluate the performance of TGNN and leading link prediction methods using PPI network structure as input. Our method (TGNN) takes interactions as edges and node degrees as node features. We compare HIGH-PPI with six heuristic methods and one DL-based method. Heuristic methods, the simple yet effective ones utilizing the heuristic node similarities as the link likelihoods, include common neighbors (CN)[33], Katz index (Katz)[50], Adamic-Adar (AA)[51], preferential attachment (PA)[52], SimRank (SR)[53] and paths of length three (L3)[22]. MLP_IP, a DL approach, learns node representations using a multilayer perceptron (MLP) and identifies the node similarity via inner product (IP) operation. We calculate the MAE and NMI values of recovered networks and highlight those with a high capacity for recovery (NMI ≥ 0.7 and MAE ≤ 0.35) in orange. Results show that link prediction methods that are more adept at recovering network properties typically perform better. This gain validates our findings in Fig. 4a and highlights the need for TGNN in the top view. In addition, a comparison of MIL_IP and L3 elucidates that pairwise learning is insufficient to well capture the network information. Although L3 can capture the evolutionary principles of PPIs to some extent, our method beats L3 by better recovering the structure of the PPI network.

We provide an example on an SHS27k sub-network. As can be seen, there exist two distinct communities connected by two inter-community edges. We use the original sub-network as inputs and find that non-TGNN link prediction methods (i.e., CN, Katz, SR, AA, PA) tend to give high scores for intercommunity interactions. As an interesting observation, when we apply the Louvain community detection algorithm[54] to the recovered structure, it cannot produce an accurate community partition as the abundant inter-community interactions disrupt the original community structure. To examine degree recovery ability, we randomly select 50% of interactions as inputs and show each method's degree recovery result for node KIF22 in Fig. 4c. We find non-TGNN approaches cannot well recover the links connecting the node KIF22 while TGNN approach can. In short, these experiments demonstrate that the structure properties of the PPI network are not always reflected in traditional link prediction methods, and moreover, capturing and learning the network structures in our top view improves the prediction performance.

## HIGH-PPI accurately identifies key residues constituting functional sites
Typically, functional sites are spatially clustered sets of residues. They control protein functions and are thus important for PPI prediction. As our proposed model has the capacity to capture spatial-biological arrangements of residues in the bottom view, this characteristic can be used to explain the model's decision. It is meaningful to notice that HIGH-PPI can automatically learn the residue importance without any residue-level annotations. In this section, we provide (1) a case study of predicting residue importance for the binding surface, (2) two cases of estimating residue importance for catalytic sites, and (3) an explainable ability comparison of precision in predicting binding sites.

First, a binding example between the query protein (PDB id: 2B6H-A) and its partner (PDB id: 2REY-A) is investigated. The ground truth binding surface is retrieved from the PDBePISA database[55], which is colored in red in Fig. 5a. Subsequently, we apply the GNN explanation approach (see Section 4.5 in "Methods" for details) on the HIGH-PPI model. As can be seen from Fig. 5a, HIGH-PPI can accurately and

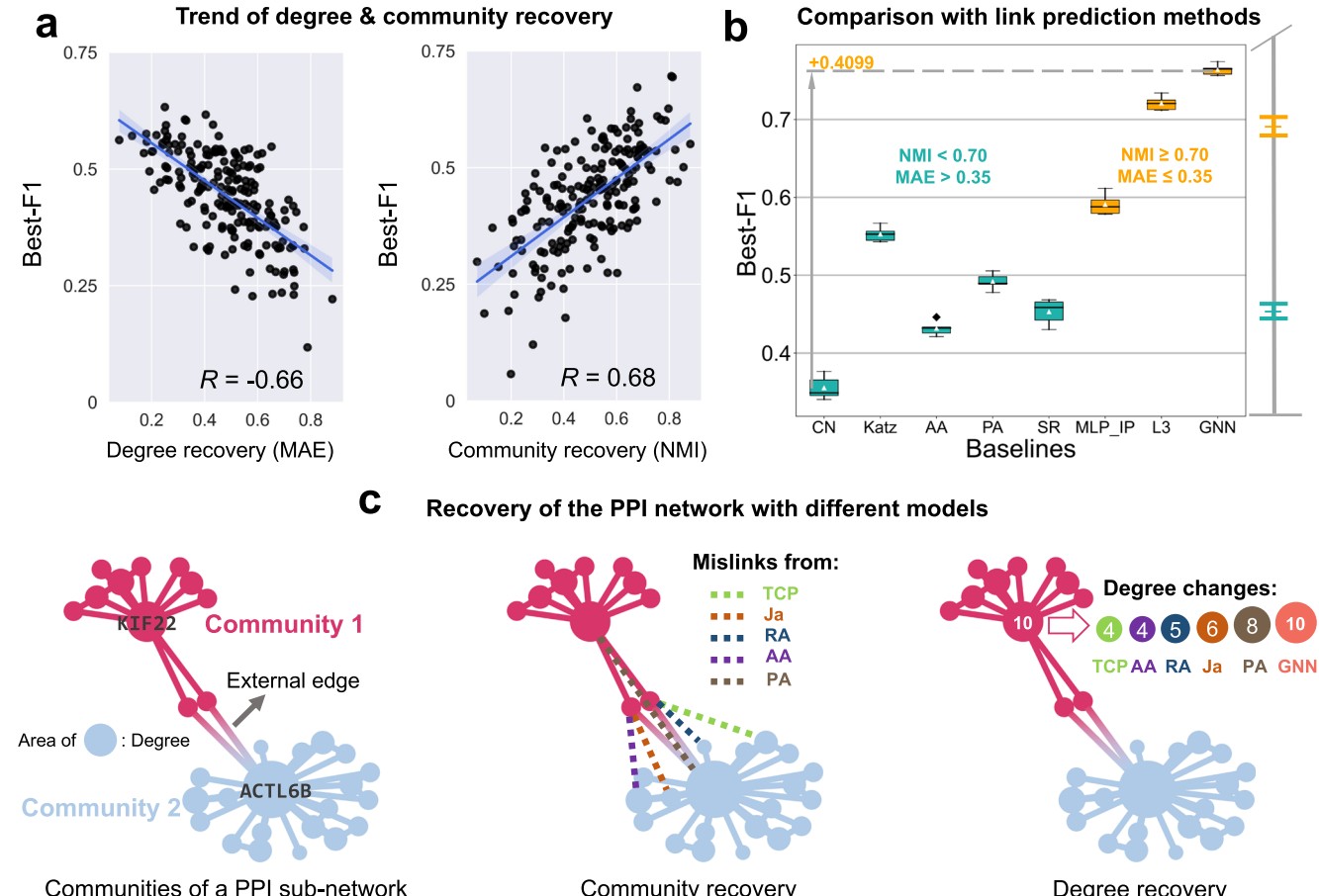

**Fig. 4 | Performance of top view GNN of HIGH-PPI to learn relational information in PPI network. a** Pearson Correlations (R) between the prediction performance (Best-F1) and degree recovery (left) and community recovery (right). It can be observed that high recovery for the degree and community of PPI network indicates better performance for PPI prediction. Degree recovery is quantified with the Mean Absolute Error (MAE) between the true and predicted degree distributions. Community recovery is quantified with the normalized mutual information (NMI) of true and predicted communities. The shaded area (error band) represents the 95% confidence interval. **b** Boxplots (center line, the median; upper and lower edges, the interquartile range; whiskers, 0.5× interquartile range) showing the Best-F1 distributions (5 runs with independent seeds) using various link prediction methods. Methods (green) predicting PPI networks of which the NMI < 0.7 and MAE > 0.35 significantly underperform the others (orange). **c** Left: An example showing a PPI network with an area of each node representing its degree value and only two external edges connecting the two communities detected. Middle: Real calculating results showing how other link prediction methods generate mislinks as external edges, which may disrupt the community partitions. Right: Real calculating results showing the disability of other link prediction methods to recover degrees. Source data are provided as a Source Data file.

automatically identify the residues belonging to the binding surface. Another observation is shown in Fig. 5c which indicates our learned residue importance is quite close to the real profiles. We show another six cases of HIGH-PPI for identifying binding surfaces correctly in Supplementary Fig. 7.

Second, in order to evaluate the prediction of catalytic sites for PPIs, we utilize the same GNN explanation approach in our model. The ground truth catalytic site is retrieved from the Catalytic Site Atlas[56] (CSA), a database for catalytic residue annotation for enzymes. We calculate the residue importance of catalytic sites for query proteins (PDB id: 1S9I-A, 1IOO-A). As seen in Fig. 5b, our proposed HIGH-PPI can correctly predict both residues for 1S9I-A and two out of three for 1IOO-A. We show another nine cases of HIGH-PPI for identifying catalytic sites in Supplementary Fig. 6, where a total of 25 out of 34 catalytic sites are correctly identified.

Additionally, we compare the model interpretability of the CNN, 3D CNN and HIGI-PPI models. We employ the CNN module in GNN-PPI[24] and 3D CNN module in DeepRank[21], respectively. We apply grad-CAM[57] and 3Dgrad-CAM[44] approaches to determine residue importance for CNN and 3D CNN models, correspondingly. We use the binding type PPIs from the STRING dataset as the training set, and randomly select 20 binding type PPIs as the test set. We use the ground truth from

PDBePISA for each query protein and treat its residues with importance >0 as surface compositions. To gauge the precision of the surface prediction, intersection over union (IoU) is used, and the box plots of the IoU score distributions are shown in Fig. 5d. The results elucidate that HIGH-PPI significantly outperforms other models in terms of interpretability with a minimum variance. In addition, 3D CNN outperforms CNN with a smaller variance, showing that 3D information supports the learning of reliable and generalized protein representations.

Protein functional site prediction sheds light on the model decisions and how to carry out additional experimental validations for PPI investigation. Excellent model interpretability also shows that our approach can accurately describe biological evidence for proteins.

## Discussion
### Hierarchical graph learning
In this paper, we study the PPI problem from a hierarchical graph perspective and develop a hierarchical graph learning model named HIGH-PPI to predict PPIs. Empirically, HIGH-PPI for PPI prediction outperforms leading methods by a significant margin. The hierarchical graph exhibits high generalization for recognizing unknown proteins and robustness against protein structure errors and PPI network perturbations.

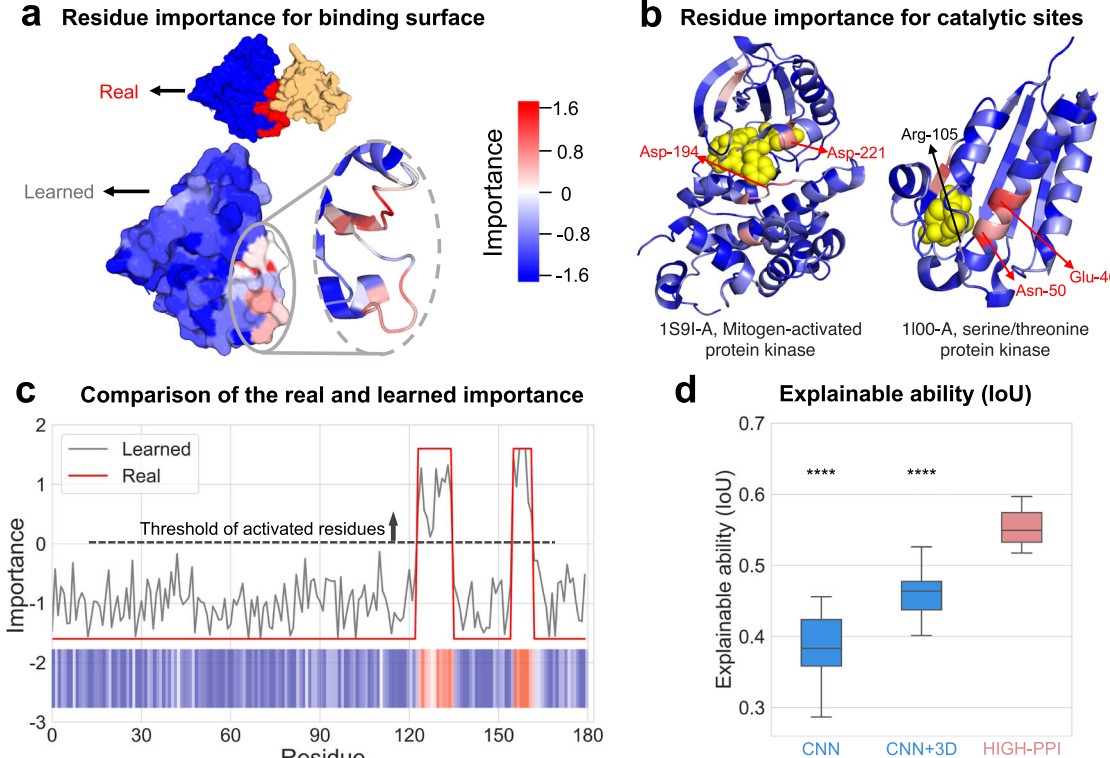

**Fig. 5 | Automatic explanation for residue importance without supervision.**
**a** Top: Depiction of a complex protein (left, query protein, PDB id: 2B6H-A; right, interacted protein, PDB id: 2REY-A) modeled in surface representation. Residues on the binding surface of query protein are highlighted in red (important) and others in blue (non-important). Bottom: Residue importance of the query protein learned from HIGH-PPI with coloring ranging from low (blue) to high (red). Important regions are magnified to show the cartoon representation. **b** Depiction of two proteins (left, PDB id: 1S9I-A; right, PDB id: 1I0O-A) modeled in cartoon representations. Residues are colored to match the importance scores, with more important residues highlighted in red and unimportant ones in blue. Residues with catalytic functions that are correctly or incorrectly identified are highlighted in red and black, respectively. **c** Polylines showing the consistency of highest peaks that represent the learned (gray) and real (red) functional regions for the binding interaction case shown in **a**. **d** Boxplots (center line, the median; upper and lower edges, the interquartile range; whiskers, $0.5 \times$ interquartile range) showing the explainable ability for binding PPIs by calculating the overlap of real and learned functional regions (IoU, Intersection over Union) with 20 PPI pairs and their real interfaces retrieved from STRING and PDBePISA database, respectively. HIGH-PPI shows greater explainable ability significantly (Two-sided $t$-test results: HIGH-PPI versus CNN ****$P = 4.4 \times 10^{-6}$, HIGH-PPI versus CNN ( +3D) ****$P = 4.4 \times 10^{-8}$). No information about residue importance was used to train our model. Source data are provided as a Source Data file.

Even without explicit supervision from binding site information, HIGH-PPI demonstrates its ability to capture residue importance for PPI with the aid of a hierarchical graph, which is a good indicator of excellent interpretability. Suppose HIGH-PPI predicts the presence of a catalytic interaction for a protein pair but identifies important sites unrelated to catalysis, we will hardly trust the model's decision. Moreover, interpretability provides trusted guides for subsequent wet experimental validations. For example, if HIGH-PPI thinks a catalytic site is important, experiments may be designed by targeting the specific site for validation.

In conclusion, interpretable, end-to-end learning with a hierarchical graph revealing the PPI nature can pave the way to map out human interactome and deepen our understanding of PPI mechanisms.

**Limitations and future work**
We describe our intuitions in the hierarchical graph learning for PPIs. The world is hierarchical. Humans tend to solve problems or learn knowledge by conceptualizing the world from a hierarchical view[58]. Due to huge semantic gaps between hierarchical views, humans always use a multi-view learning strategy to deepen the understanding of one view from the other one. Given rich hierarchical information, recent machine intelligence methods can effectively learn knowledge in each separate view but are not experts in gaining mutual benefits from both views. This is the challenge that our hierarchical world presents to

machine intelligence. Here we connect both views by employing the forward and backward propagation of DL models. The forward propagation benefits the learning for the PPI network in the top view. In turn, the backward propagation optimizes the PPI-appropriate protein representations in the bottom view.

We describe two main limitations of HIGH-PPI and outline potential solutions in future work. (1) We did not explore in depth how to use protein-level annotations. Annotations for protein functions are becoming more available due to the recent growth of protein function databases (e.g., the UniProt Knowledge-base[59]) and computational methods[29] for protein function prediction. Some annotations may speed up learning PPIs. For example, two proteins with low scores of the "protein binding" function term hardly interact with each other. We suggest that future work may consider leveraging function annotations to enhance the expressiveness of protein representations. Inspired by the contrastive learning principle, a potentially feasible solution is to enhance the consistency in protein representations and functions. (2) Protein domain information may be beneficial for hierarchical models. We clarify the core ideas here and provide a detailed description in Supplementary Method 2. Domains are distinct functional or structural units in proteins and are responsible for PPIs and specific protein functions. Both in terms of structures and functions, the protein domain can represent a crucial middle scale for the PPI hierarchy. However, to our knowledge, true (native) domain annotations are not easily available and predicted ones are usually retrieved

from computational tools, which inevitably leads to data unreliability. If we employ the domain scale as a separate view, data unreliability may spread to other views and impair the entire hierarchical model. On this basis, we prefer to recommend domain annotations as supervised information at the residue level. Precisely, a well-designed regularization is required to guarantee that all functional sites, discovered by HIGH-PPI, belong in the prepared domain database. The domain regularization and the PPI prediction loss form a flexible trade-off of learning objectives, which can appropriately tolerate the domain annotation unreliability. (3) Memory requirement grows with the view number of a hierarchical graph. HIGH-PPI employs two views to form the hierarchical graph and treat amino acid residues as microscopic components of proteins. However, we did not further consider one more microscopic view where atoms, the components of residues, provide information for representing residues. It might be beneficial to introduce an atom-level view and develop a memory-efficient way for storing and processing explicit 3D atom-level information. (4) In future work, model robustness can be further improved. Although our model outperforms in the robustness evaluation (see Supplementary Table 3), we observe that $FDR_{pre}$ is most impacted by unreliable data, which is mostly because the number of FP significantly increases (up to 6 times) from Data 1 to Data 9. A possible explanation for the significant rise in FP is that the model's "low demand" for a positive sample permits certain controversial samples to be projected as true. To address this issue, we recommend the future work consider a straightforward method—the voting strategy which uses the voting outcomes of various independent classifiers to identify true PPIs. Independence makes it unlikely for voting classifiers to commit the same errors. A test pair can only be predicted as true if it is approved by most voting classifiers, which makes the model more demanding for the PPI presence.

## Methods

### Construction of a hierarchical graph

We denote a set of amino acid residues in a protein as $Prot = \{r_1, r_2, \ldots, r_n\}$. Each residue is described with $\theta$ kinds of physicochemical properties. For the bottom inside-of-protein view, a protein graph $g_b = (V_b, A_b, X_b)$ is constructed to model the relationship between residues in $Prot$, where $V_b \subseteq Prot$ is the set of nodes, $A_b$ is an $n \times n$ adjacency matrix representing the connectivity in $g_b$, and $X_b \in \mathbb{R}^{n \times \theta}$ is a feature matrix containing the properties of all residues.

For the top outside-of-protein view, a set of protein graphs can be interconnected within a PPI graph $g_t$, which is denoted as $g_b \in V_t$. The connectivity (i.e., interactions) between protein graphs can be denoted as an $m \times m$ adjacency matrix $A_t$. In addition, $X_t \in \mathbb{R}^{m \times \varnothing}$ represents a feature matrix containing the representations of all proteins. We model the protein graphs and their connections as a hierarchical graph, in which four key variables (i.e., $A_b$, $X_b$, $A_t$, $X_t$) need to be clarified.

(1) The adjacency matrix $A_b \in \{0,1\}^{n \times n}$ in the protein graph and protein contact map are exactly equivalent. Contact maps are obtained with atomic level 3D coordinates of proteins. First, we retrieve the native protein structures from the Protein Data Bank[60] and protein structures of various RMSD scores by AlphaFold[43]. Then we represent the location of each residue by the 3D coordinate of its $C_\alpha$ atom. The presence or the absence of contact between a pair of residues is decided by their $C_\alpha - C_\alpha$ physical distance. We perform a sensitivity analysis (see Supplementary Fig. 8) and find that our model produces similar results when trained on contact maps with cutoff distances ranging between 9 Å to 12 Å. Finally, we choose the optimal cutoff distance of 10 Å, which allows our model to peak its performance. (2) For a feature matrix $X_b$, each row represents a set of properties for one amino acid residue. In this work, seven residue-level properties are considered (i.e., $\theta = 7$): isoelectric point, polarity, acidity and alkalinity, hydrogen bond acceptor, hydrogen bond donor, octanol-water

partition coefficient, and topological polar surface area. Supplementary Data File 3 contains quantitative values of seven types of properties for each amino acid. All properties can be easily retrieved from the RDKit repository[61]. (3) The PPI network structure determines the adjacency matrix $A_t \in \{0,1\}^{m \times m}$, in which the $i$-th row and $j$-th column element is 1 if the $i$-th and $j$-th proteins interact. (4) The $i$-th row of the feature matrix $X_t$ represents the representation vector for the $i$-th protein graph $g_b$.

### BGNN for learning protein representations

We use the bottom view graph neural networks (BGNN) to learn protein representations. Graph convolutional networks (GCNs) have shown great effectiveness for relational data and are suitable for learning graph-structured protein representations. Thus, we propose BGNN based on GCNs.

Given the adjacency matrix $A_b \in \{0,1\}^{n \times n}$ and the feature matrix $X_b \in \mathbb{R}^{n \times \theta}$ of an arbitrary protein graph $g_b$, BGNN outputs the residue-level representations in the first GCN block, $H^{(1)} \in \mathbb{R}^{n \times d_1}$:

$$H^{(1)} = GCN(A_b, X_b) \tag{1}$$

where $d_1$ is the embedding dimension for the first GCN layer.

Formally, we update residue representations with the neighbor aggregations based on the work of Kipf and Welling[36]:

$$H^{(1)} = BN\left(ReLU\left(\widetilde{D}^{-1/2}(A_b + I_n)\widetilde{D}^{-1/2}X_b W^{(1)}\right)\right) \tag{2}$$

where $I_n \in \mathbb{R}^{n \times n}$ is the identity matrix, $\widetilde{D} \in \mathbb{R}^{n \times n}$ is the diagonal degree matrix with entries $D_{ii} = \sum_j (A_b + I_n)_{ij}$, $W^{(1)} \in \mathbb{R}^{\theta \times d_1}$ is a learnable weight matrix for the GCN layer, ReLU, BN denotes the ReLU activation function and batch normalization, respectively.

With the learnable weight matrix $W^{(2)} \in \mathbb{R}^{d_1 \times d_2}$, the second GCN block produces the output $H^{(2)} \in \mathbb{R}^{n \times d_2}$:

$$H^{(2)} = BN\left(ReLU\left(\widetilde{D}^{-1/2}(A_b + I_n)\widetilde{D}^{-1/2}H^{(1)} W^{(2)}\right)\right) \tag{3}$$

Finally, we perform the readout operation with a self-attention graph pooling layer[39] and average aggregation to obtain the entire graph representation of a fixed size, $x \in \mathbb{R}^{1 \times d_2}$. To clarify, we use $x_i \in \mathbb{R}^{1 \times d_2}$ to represent the final representation for the $i$-th protein graph.

### TGNN for learning PPI network information

We use the top view graph neural networks (TGNN) to learn PPI network information. We are inspired by graph isomorphism network (GIN[37]), which has the superb expressive power to capture graph structures. Formally, we are given the PPI graph $g_t = (V_t, A_t, X_t)$, where $X_t \in \mathbb{R}^{m \times d_2}$ is defined as the feature matrix whose row vector is a final protein representation from BGNN (i.e., $X_t^{[i,:]} = x_i, i = 1, 2, \ldots, m$). TGNN updates the representation of protein $v$ in the $k$-th GIN block:

$$x_v^{(k)} = BN\left(ReLU\left(MLP^{(k)}\left((1+\epsilon) \bullet x_v^{(k-1)} + \sum_{u \in \mathcal{N}(v)} x_u^{(k-1)}\right)\right)\right) \tag{4}$$

where $x_v^{(k)}$ denotes the representation of protein $v$ after the $k$-th GIN block, $\mathcal{N}(v)$ is a set of proteins adjacent to $v$, and $\epsilon$ is a learnable parameter. We denote the inputs of protein representations for the first GIN block as $x_i^{(0)} = x_i, i = 1, 2, \ldots, m$.

After three GIN blocks, TGNN produces representations for all proteins. For an arbitrary query pair containing the $i$-th and $j$-th proteins, we use the concatenation operation to combine the representations of $x_i^{(3)}$ and $x_j^{(3)}$. A fully connected layer (FC) is employed as the classifier. The final vector $\hat{y}_{ij} \in \mathbb{R}^{1 \times c}$ for the presence probability of PPI is denoted as $\hat{y}_{ij} = FC\left(h_i^{(3)} \| h_j^{(3)}\right)$ where $c$ denotes the total number of PPI types involved and $\|$ denotes the concatenation operation.

## Model training details

Given a training set $\mathcal{X}_{train}$ and ground truth labels for multi-type PPIs $\mathcal{Y}_{train}$, we train BGNN and TGNN in an end-to-end manner by minimizing the loss function of multi-task binary cross-entropy:

$$\mathcal{L}(\Theta) = \sum_{k=0}^{c} \left( \sum_{x_{ij} \in \mathcal{X}_{train}} -y_{ij}^k \log \hat{y}_{ij}^k - \left(1 - y_{ij}^k\right) \log\left(1 - \hat{y}_{ij}^k\right) \right) \quad (5)$$

where $\Theta$ is the set of all learnable parameters, and $ij$ denotes the ground truth of the $k$-th type PPI of the $i$-th and $j$-th proteins.

We determine all the hyper-parameters through a grid search based on a 5-fold cross-validation. For BGNN, we set the output dimension $d_1, d_2$ of weight matrix to 128. For each GIN block in TGNN, we use a two-layer MLP and set the output dimension of each layer to 64. As the STRING dataset contains seven types of PPIs, we set the output dimension of the FC layer to $c = 7$. We use the Adam optimizer with a learning rate $lr = 0.001$, $\beta_1 = 0.99$, $\beta_2 = 0.99$, a batch size of 128, and the default epoch number of 500. We train all of the model parameters until convergence in each cross-validation.

## Residue importance computation

We employ the method called GNNExplainer[62] to generate explanations for HIGH-PPI. By taking the well-trained GNN model and its predictions as inputs, GNNExplainer returns the most important subgraph by maximizing the mutual information $MI$ between the model prediction and possible subgraphs. Motivated by this, we directly formalize the notion of subgraph importance using $MI$ and further compute the importance of all nodes (i.e., residues).

Given protein graphs $G_1$ and $G_2$ that connect in the PPI network, our goal is to identify the node importance of $G_1$. According to GNNExplainer, once sampling a random subgraph $G_s \subseteq G_1$, we obtain the entire importance of $G_s$ as follow:

$$MI_s(Y, G_s) = H(Y) - H(Y|G = G_s) \quad (6)$$

where $MI_s$ represents importance of $G_s$, $Y$ is a variable indicating the probability of PPI presence of $G_1$ and $G_2$, and $H(\bullet)$ is the entropy term.

Assume that all nodes in the subgraph $G_s$ contribute equally to the $MI$ value, we obtain the batch importance for each node in $G_s$. The final importance score for a specific node is the average of all its batch importance scores. For example, if a node $v$ contributes 0.4 and 0.6 for two sampled subgraphs respectively, the final importance of node $v$ is 0.5. To facilitate comparison, we compute the $z$-scores of final residue importance for standardization:

$$z_s = \frac{z_f - \mu}{\sigma} \quad (7)$$

where $z_f \in \mathbb{R}^{1 \times n}$ is the finally computed importance vector for all residues, $\mu$ is the average of $z_f$, $\mu$ is the standard deviation of $z_f$, and $z_s \in \mathbb{R}^{1 \times n}$ is the $z$-score importance after standardization.

## Statistics and reproducibility

As indicated in figure legends, data in bar charts are represented as mean ± standard deviation (SD). For all boxplots, the center line represents the median, upper and lower edges represent the interquartile range, and the whiskers represent 0.5× interquartile range. The statistical significance between the two groups was obtained by a two-sided $t$-test with $P$-value < 0.05 considered significant.

## Reporting summary

Further information on research design is available in the Nature Portfolio Reporting Summary linked to this article.

## Data availability

The PPI and protein data used in this study are available in the Zenodo database under "Accession Code 7213401". They are obtained from the following publicly available database. Datasets containing protein sequences and their interaction annotations are obtained from https://github.com/muhaochen/seq_ppi. The native protein structures are obtained from PDB: https://www.rcsb.org/. Protein structures with errors are obtained from AlphaFold: https://alphafold.ebi.ac.uk/. The catalytic site information of proteins can be found at CSA: https://www.ebi.ac.uk/thornton-srv/m-csa/. The ground truth of binding site information is obtained from PDBePISA: https://www.ebi.ac.uk/pdbe/pisa/. All other relevant data supporting the key findings of this study are available within the article and its Supplementary Information files or from the corresponding author upon reasonable request. Source data are provided with this paper.

## Code availability

An open-source software implementation of HIGH-PPI is available at https://github.com/zqgao22/HIGH-PPI. The source code can be cited by using https://doi.org/10.5281/zenodo.7600622.

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

## Acknowledgements

The research of Li was supported by National Natural Science Foundation of China (Grant No. 62206067), Tencent AI Lab Rhino-Bird Focused Research Program RBFR2022008 and Guangzhou-HKUST(GZ) Joint Funding Scheme. The research of Huang was supported by the National Natural Science Foundation of China (Grant No. 21825101).

## Author contributions

Z.G. and C.J. wrote the first draft of manuscript. J.L., Y.H., and L.L. revised the manuscript to the submitted version. J.L., Z.G., Y.H., and H.Y. conceived the study. Z.G. designed all the experiments and wrote the codebase of HIGH-PPI. Z.G., J.Z., and X.J. conduct the benchmarks, and run all of the analysis. X.J. collected and preprocessed protein contact maps. Z.G., L.L., and P.Z. contributed to data analysis and model discussion. J.Z. conducted the figure design for overall framework. Z.G., C.J., J.Z., and X.J. completed the visualizations. J.L. and Y.H. supervised the research. All of the authors reviewed the manuscript and approved it for submission.

## Competing interests

The authors declare no competing interests.
