## [Peer Review File · Nature Communications]

Hierarchical Graph Learning for Protein-Protein InteractionReviewer #1 (Remarks to the Author):

The paper titled "Hierarchical Graph Learning for Protein-Protein Interaction" proposed a new computational method for prediction protein-protein interactions. This method integrated the protein-protein interaction network and protein 3D structure information to achieve good performances. The authors proposed that these two kinds of information can be integrated together by hierarchical graph model. Graph neural network-based methods can be developed on the hierarchical graph. Particularly, the authors proposed two views of protein-protein interaction, the bottom inside-of-protein view and the top outside-of-protein view. The two views described protein-protein interactions at two different scale, the residue-residue intra-molecular scale and the protein-protein inter-molecular scale. This kind of information integration is new under the topic of protein-protein interaction. The application of graph neural network, or more precisely GCN and GIN is good with sophisticated designs, achieving an end-to-end predictive model. The validation results of the method show that it has better prediction performance, good robustness and good generalization ability. I think this is an innovative contribution to the bioinformatics field with great potential. I have several minor comments, which I would like the authors to clarify or just verify in the revisions.

(1) I believe that the authors knew that the quality of high-throughput protein-protein interaction data is always a problem in predicting protein-protein interaction. This problem has two sides. One is that the positives may contain a considerable number of false positives. The other is that the negatives are not reliable. Both sides affect machine learning models. The authors applied, "perturbations including random addition or removal to known interactions", in order to prove the robustness of their methods. This is good. However, robust models may bring in false discovery in practical applications. I think it would be better if the authors can perform some randomization or permutation tests to prove the statistical quality of their method. If the authors feel this is of too much work or not necessary, they may try to make some discussions.

(2) In Fig3(d), the authors analyzed the importance of different properties in their model. I am not sure whether this proves their choices of the seven physicochemical properties. There are about 1/3 "blocks" in a very light color, indicating a near zero z-score. The authors may need to make more discussions on why and how they choose the seven physicochemical properties, but not others.

(3) In the discussions, the authors mentioned that a future work may be considered at atomic level. I think there is a middle-scale view, which is at domain-scale. Maybe this can be discussed. Surely, if the authors feel this is unnecessary or not proper in this paper, they may choose to ignore this.

(4) Although the authors have shared the data and code, I still have no time to validate/reproduce the results myself. However, I managed to browse the data and have a suggestion. The authors should share the data in pure text format, like csv/tsv/txt etc., but not the numpy/npz format, which is not convenient for quick browsing.

(5) There are several minor confusing writings in the text and math, I think the authors should verify them. I also suggest the authors to proof read the manuscript again, as some kind of "hurry" can be felt when reading its current version.

On Page 6, legends of the fig1 "SEPRINA3" should be "SERPINA3"? Please notice the spelling

On Page 18, Line 332, I think it is "3.2", not "3.1"

On Page 20, Line 363, "gb \in gt", this does not make sense. I think it should be "gb \in Vt", according to the words on Line 362

On Page 23, Line 433, "G1 \subseteq G1", this does not make sense, please verify.

Reviewer #2 (Remarks to the Author):

Comments to the Author:

The manuscript 'Hierarchical Graph Learning for Protein-Protein Interaction' submitted by Gao et al

deals with a novel hierarchical graph learning based method allowing the authors to predict interactions for given protein pairs and key residues for their interactions. The authors used multi-type human protein-protein interactions (PPI) and showed that their method demonstrates high accuracy and robustness in predicting PPIs. At the core of their method is a synergistic predictive effect of two levels: the level of the structural representations of the protein itself (lower level) and the level of the PPI network (upper level). By mutually sharing the information learned in each of the lower and upper levels, it leads to better structural representations of proteins and learning relationships between proteins. Their method is novel in that it models the natural PPI relationship by hierarchically linking both levels. In addition, it is also interesting to graphically represent protein and network structures using Graph Neural Networks (GNN) for this modelling. Furthermore, the authors have rigorously evaluated the robustness of their method from various viewpoints, increasing the reliability of their method. The manuscript is well written and well-illustrated. I think that it is worthy of being published in Nature Communication. I only have some minor questions and comments:

Minor questions and comments:

- Page 7, in the section 2.1, line 134-136: The proposed method is trained and evaluated using a dataset consisting of 1,690 proteins and 7,624 PPIs. By simple calculation, one protein is involved in 4.5 PPIs ($7,624 \text{ PPIs} / 1,690 \text{ proteins} = 4.511\dots$). According to Park and Marcotte (Nat Methods. 2012 December; 9(12): 1134–1136. doi:10.1038/nmeth.2259.), PPI prediction methods tend to perform much better on test pairs that share the same proteins with the training dataset than those that do not. Although the generalization evaluation (OOD) has been done, it would be useful for readers to show the overall performance and robustness evaluation in C2 and C3 among 3 distinct classes proposed by Park and Marcotte.
- Page 20, in the section 4.1, line 370-372: The presence or the absence of contact between a pair of residues is determined by their C_α-C_α physical distance, and the threshold is defined as ~10 Å. Please explain how you defined this threshold value. Is this value (~10 Å) a reasonable distance?
- Page 10, in the section 2.3, line 177: "BFS-0.3 denotes that the test set involving 30% known proteins ..." should be "BFS-0.3 denotes that the test set involves 30% known proteins ...".
- Page 12, in the section 2.3, line 199: "GNNPPI" should be "GNN-PPI".
- Page 20, in the section 4.1, line 366: Is "(i.e., A_b, X_b, A_t, A_t)" a mistake of "(i.e., A_b, X_b, A_t, X_t)"?
- Page 23, in the section 4.5, line 433: Is "G₁⊆G₁" a mistake of "G_s⊆G₁"?

Point-by-point responses to reviewer comments:

NCOMMS-22-42373 "Hierarchical Graph Learning for Protein-Protein Interaction" by Ziqi Gao, et al.

We are very grateful to the Editor and Reviewers for their critical comments on the original manuscript. We have now addressed in detail all your concerns, which, we think, has greatly improved the quality and readability of our paper.

Please find the point-by-point responses (in black) to reviewer comments (in blue). All edits have been colored in red in our revised manuscript version and the revised supplementary information.

Reviewer #1 (Remarks to the Author):

The paper titled "Hierarchical Graph Learning for Protein-Protein Interaction" proposed a new computational method for prediction protein-protein interactions. This method integrated the protein-protein interaction network and protein 3D structure information to achieve good performances. The authors proposed that these two kinds of information can be integrated together by hierarchical graph model. Graph neural network-based methods can be developed on the hierarchical graph. Particularly, the authors proposed two views of protein-protein interaction, the bottom inside-of-protein view and the top outside-of-protein view. The two views described protein-protein interactions at two different scale, the residue-residue intra-molecular scale and the protein-protein inter-molecular scale. This kind of information integration is new under the topic of protein-protein interaction. The application of graph neural network, or more precisely GCN and GIN is good with sophisticated designs, achieving an end-to-end predictive model. The validation results of the method show that it has better prediction performance, good robustness and good generalization ability. I think this is an innovative contribution to the bioinformatics field with great potential. I have several minor comments, which I would like the authors to clarify or just verify in the revisions.

We want to thank you for your appreciation of our effort and for highly recognizing that HIGH-PPI may offer a lot of potential for the bioinformatics community. To address your concerns, we provide point-by-point responses as below.

(1) I believe that the authors knew that the quality of high-throughput protein-protein interaction data is always a problem in predicting protein-protein interaction. This problem has two sides. One is that the positives may contain a considerable number of false positives. The other is that the negatives are not reliable. Both sides affect machine learning models. The authors applied, "perturbations including random addition or removal to known interactions", in order to prove the robustness of their methods. This is good. However, robust models may bring in false discovery in practical applications. I think it would be better if the authors can perform some randomization or permutation tests to prove the statistical quality of their method. If the authors feel this is of too much work or not necessary, they may try to make some discussions.

Thank you for the question on the robustness evaluation in our paper and for suggesting a more

systematic evaluation protocol. We agree with you that it is necessary and valuable to investigate the statistical quality of false positive (FP) and false negative (FN) performance, particularly for the protein-protein interaction (PPI) data that may not be reliable. To address this issue, we have made some discussions of our robustness evaluation experiment (Fig. 2b) and conducted an additional experiment to provide evidence.

On one hand, we respectfully clarify the intuition of the robustness evaluation (Fig. 2b) in our manuscript. We collectively refer to the potential FP and FN samples as **unreliability**. In essence, both the training set and the test set conceal **unreliability** in the involved dataset (*e.g.*, SHS27k). The most ideal robustness evaluation requires us to keep the training set containing **unreliability** unchanged and remove the unreliable labels from the test set, but this is unfortunately an intractable problem. Thus, in order to conduct the robustness evaluation experiments, we made a compromise that **we consider the original dataset to be reliable and artificially add perturbations to represent data unreliability**. Importantly, perturbations are only added to the training set labels, while the test set labels remain unchanged (reliable). This simulates the fact that the PPI data we use in the training model is inherently unreliable, while we evaluate the trained model with its predictions and reliable test set labels.

Dataset	Data 1	Data 2	Data 3	Data 4	Data 5	Data 6	Data 7	Data 8	Data 9
FPR_{train} / FNR_{train}	0.1/0.1	0.2/0.2	0.3/0.3	0.4/0.4	0.5/0.5	0.6/0.6	0.7/0.7	0.8/0.8	0.9/0.9

Table R1. Statistics of FPR_{train} and FNR_{train} of 9 datasets created for training the models.

On the other hand, to address your concern, we designed the following experimental protocol, and compared the statistical performance of our approach HIGH-PPI with a solid baseline (*i.e.*, GNN-PPI). In brief, we want to explore whether the unreliability of the data causes the unreliability of our model. First, we assume that the original SHS27k dataset has almost no FP or FN PPI samples. We name the original dataset as **reliable data** and dataset containing FP or FN PPI samples as **unreliable data**. Second, we create 9 unreliable datasets with various FP rates (FPR_{train}) and FN rates (FNR_{train}) shown in Table R1. Then, we report robustness evaluation metrics including FPR , FNR and false discovery rate (FDR). Note that herein we use FPR_{train} and FNR_{train} to represent the unreliability of **input data for training the model**, and use FPR_{pre} , FNR_{pre} and FDR_{pre} to evaluate the robustness for **model’s prediction**.

$$\begin{aligned}
 FPR_{pre} &= \frac{FP}{FP+TN} \\
 FNR_{pre} &= \frac{FN}{TP+FN} \\
 FDR_{pre} &= \frac{FP}{FP+TP}
 \end{aligned}
 \tag{R1}$$

Finally, we estimate the statistical quality of each metric with **permutation test (PT)**, after obtaining FPR_{pre} , FNR_{pre} and FDR_{pre} scores from testing HIGH-PPI and GNN-PPI that are trained on

the above mentioned 9 datasets in Table R1. The computation algorithm of PT in terms of a single metric is summarized in following pseudo-code.

Computation algorithm for **permutation test**

Input: e scores (e.g., FPR_{pre}) of HIGH-PPI ($x_H \in \mathbb{R}^{1 \times e}$) and GNN-PPI ($x_G \in \mathbb{R}^{1 \times e}$).

Parameter: Null hypothesis ($H(0)$: x_H is not significantly different from x_G), permutation time ($\mathcal{N} = 10000$), confidence ($\mathcal{C} = 99\%$), permutation distribution ($\mathcal{D} \in \mathbb{R}^{1 \times \mathcal{N}}$).

Output: The decision for null hypothesis pt .

```

1 Calculate the observed statistics  $t_{obs} = mean(x_H) - mean(x_G)$ .
2  $x_{all} \in \mathbb{R}^{1 \times 2e} \leftarrow Concatenate(x_H, x_G)$ 
3 for  $i = 1$  to  $\mathcal{N}$  do
4    $x_{all}^i \leftarrow$  random reordering of  $x_{all}$ 
5    $x_H^i \leftarrow x_{all}^i[:e]$ .
6    $x_G^i \leftarrow x_{all}^i[-e:]$ .
7    $\mathcal{D}_i \leftarrow mean(x_H^i) - mean(x_G^i)$ 
8 Calculate P-value  $p_{value} \leftarrow \#(\mathcal{D} > t_{obs})/\mathcal{N}$ 
9 if  $p_{value} < 1 - \mathcal{C}$  then
10    $pt = 'H(0) Rejected'$ 
11 else
12    $pt = 'H(0) Accepted'$ 
13 Return  $pt$ 

```

- Experiment 1: We train the model on the reliable dataset and calculate FPR_{pre} , FNR_{pre} and FDR_{pre} scores with reliable test set labels. Finally, we obtain the performance significance of those three metrics with permutation tests.
- Experiment 2: We train the model on the created 9 unreliable datasets and calculate FPR_{pre} , FNR_{pre} and FDR_{pre} scores with reliable test set labels. Finally, we obtain the performance significance of those three metrics with permutation tests.

Results:

Seeds		1	2	3	4	5	6	7	8	p_{value}	pt
FPR	BI	0.499	0.490	0.529	0.549	0.546	0.516	0.532	0.527	0.03842	NS
	Ours	0.530	0.517	0.501	0.482	0.487	0.506	0.495	0.494		
FNR	BI	0.200	0.187	0.181	0.177	0.191	0.181	0.181	0.194	0.00012	S
	Ours	0.132	0.130	0.138	0.142	0.115	0.119	0.140	0.153		
FDR	BI	0.199	0.181	0.200	0.207	0.221	0.190	0.201	0.211	0.00015	S
	Ours	0.147	0.138	0.138	0.134	0.110	0.122	0.138	0.150		

Table R2. Results of Experiment 1. We show the results of three metrics after running 8 seeds and obtain the p_{value} and permutation test decision pt , where ‘S’ represents significant performance and ‘NS’ represents the performance is not significant. Besides, ‘BI’ and ‘Ours’ represent the solid baseline method (GNN-PPI) and HIGH-PPI, respectively. A highlighted ‘Ours’ represents HIGH-PPI model achieves better performance in terms of the average score across 8 seeds.

Unreliable data		Data 1	Data 2	Data 3	Data 4	Data 5	Data 6	Data 7	Data 8	Data 9	Δ	p_{value}	pt
FPR	BI	0.537	0.570	0.581	0.604	0.620	0.602	0.657	0.739	0.717	0.180	0.00004	S
	Ours	0.505	0.526	0.545	0.555	0.563	0.575	0.621	0.655	0.665	0.160		
FNR	BI	0.192	0.266	0.306	0.348	0.368	0.422	0.440	0.565	0.698	0.506	0.00410	S
	Ours	0.171	0.180	0.199	0.213	0.236	0.262	0.362	0.599	0.641	0.470		
FDR	BI	0.216	0.324	0.380	0.449	0.487	0.524	0.600	0.787	0.854	0.638	0.00032	S
	Ours	0.174	0.196	0.229	0.252	0.285	0.324	0.482	0.739	0.780	0.606		

Table R3. Results of Experiment 2. We show the results of three metrics after testing the model on reliable test set labels and obtain the p_{value} and permutation test decision pt , where ‘S’ represents significant performance and ‘NS’ represents the performance is not significant. Besides, ‘BI’ and ‘Ours’ represent the methods GNN-PPI and HIGH-PPI, respectively. A highlighted ‘Ours’ represents the HIGH-PPI model achieves better performance in terms of the average score across 9 unreliable datasets. Δ represents the difference between the maximum and minimum value.

Results and Discussions:

Experiment 1 (Table R2) mainly evaluates the unreliability of predictions of models that are trained with reliable data. We observe that HIGH-PPI achieves better performance (*i.e.*, average results) of each metric, with different statistical qualities calculated with permutation tests. More precisely, our model shows insignificant superiority ($p_{value} = 0.03842$) in the FPR_{pre} metric, and considerable superiority in the FNR_{pre} ($p_{value} = 0.00012$) and FDR_{pre} ($p_{value} = 0.00015$) metrics, which illustrates the performance superiority of our model and leads to the following more in-depth analysis.

The AI-assisted PPI prediction's primary goal is to effectively discover or choose possible protein partners for further validation. Determining "more real PPIs are selected by AI models (*i.e.*, TP or $Recall$)" rather than "more PPIs selected by AI models are real" is, in our opinion, of greater significance. In this regard, we are glad to find that our model significantly outperforms the competition, particularly when it comes to the TP -related metrics (*i.e.*, FNR_{pre} , FDR_{pre}). We sincerely hope that future work can optimize all robustness metrics in a comprehensive manner.

Experiment 2 (Table R3) mainly evaluates the model robustness against unreliable training data. We can see that even when our proposed model encounters unreliable data, it still performs better (*i.e.*, average results) of all three metrics. Surprisingly, while retaining the original significance in FNR_{pre} and FDR_{pre} , our model substantially improves the superiority significance in the FPR_{pre} metrics ($p_{value}: 0.03842 \rightarrow 0.00004$). This displays good statistical quality of our model in terms of the robustness evaluation. Moreover, we note that FDR_{pre} is most impacted by unreliable data (bigger Δ), which is mostly because the number of false positives (FP) significantly increases (up to 6 times) from Data 1 to Data 9.

As a result, the model appears to be more sensitive to the FPR_{train} and filters extensive protein pairs that are not genuinely true, even though the FPR_{train} and FNR_{train} in each unreliable dataset are the same. Simply put, the difficulty of the follow-up validation will directly correlate with the model's robustness, therefore unreliable data result in the expense of more time-consuming follow-up experimental validation. To address this issue, we instinctively suggest a straightforward potential option for further research. A possible explanation for the rise in FP is that the model's "low demand" for a positive sample permits certain controversial samples to be projected as true. In light of this, we recommend future research to use a voting strategy which uses the voting outcomes of various independent classifiers to identify true PPIs. Independence makes it unlikely for voting classifiers to commit the same errors, increasing the need of the model for correctly predicting samples—to be approved by the majority of voting classifiers.

Supplementary notes:

1. We have stored more detailed results (*e.g.*, specific values for TP) in a csv file (named `revision_1.csv`) here so that you can view them quickly and conveniently.
2. We have revised the manuscript according to this comment. Specifically, we have added experimental analysis on robustness evaluation in the third paragraph of Section 2.2 (Line 174-185).

“Furthermore, we perform false discovery on our method, which investigates the effect of the

training data unreliability (*i.e.*, false negative (*FN*) and false positive (*FP*)) on our model and a solid baseline (GNN-PPI). Specifically, we consider the original dataset to be reliable and artificially add perturbations to represent data unreliability. Supplementary Table 1 shows the created 9 datasets with different *FP* rates (FPR_{train}) and *FN* rates (FNR_{train}). We respectively train the model on the reliable training set and created 9 unreliable ones and present the *FP* rates (FPR_{pre}), *FN* rates (FNR_{pre}) and false discovery rates (FDR_{pre}) metrics on the test sets (see Supplementary Table 2 and 3). Without unreliability, our model achieves best performance with insignificant superiority ($*P = 3.8 \times 10^{-2}$) in the FPR_{pre} metric, and considerable superiority in the FNR_{pre} ($***P = 1.2 \times 10^{-4}$) and FDR_{pre} ($***P = 1.5 \times 10^{-4}$) metrics. When introducing data unreliability, we are surprised to find that our model substantially improves the superiority significance in the FPR_{pre} metric ($****P = 4.0 \times 10^{-5}$) while retaining the original significance in FNR_{pre} and FDR_{pre} . In addition to showing the excellent robustness of our model, we also provide more in-depth insights in Section 3.2.”

3. We have added results for robustness evaluation experiments in the table form (Supplementary Table 2, 3) to the Supplementary Information file.
4. We have added the comments for future work on improving model robustness in Section 3.2 (Line 393-402).

“(4) In the future work, model robustness can be further improved. Although our model outperforms in the robustness evaluation (see Supplementary Table 3), we observe that FDR_{pre} is most impacted by unreliable data, which is mostly because the number of *FP* significantly increases (up to 6 times) from Data 1 to Data 9. A possible explanation for the significant rise in *FP* is that the model's "low demand" for a positive sample permits certain controversial samples to be projected as true. To address this issue, we recommend the future work to consider a straightforward method—the voting strategy which uses the voting outcomes of various independent classifiers to identify true PPIs. Independence makes it unlikely for voting classifiers to commit the same errors. A test pair can only be predicted as true if it is approved by most voting classifiers, which makes the model more demanding for the PPI presence.”

(2) In Fig3(d), the authors analyzed the importance of different properties in their model. I am not sure whether this proves their choices of the seven physicochemical properties. There are about 1/3 "blocks" in a very light color, indicating a near zero z-score. The authors may need to make more discussions on why and how they choose the seven physicochemical properties, but not others.

Thanks for your insightful comments. We strongly agree with you that more discussion about physicochemical properties selection is necessary in our work. We respectfully clarify that feature importance exploration in the manuscript (Fig. 3d) is intended to better explain our model and explore potential biomarkers of PPIs. However, we realize it to be ambiguous in the manuscript and we apologize for the ambiguity in the description of feature importance. In fact, the 7 features shown in Fig. 3d are chosen ones from 12 optional features (Table R4), and these **7 important features** constitute the optimal set that enables the model to operate at its peak. ‘A near zero z-score’ represents a relative importance among those important features in Fig. 3d.

Then, to address your question about “how they choose the seven physicochemical properties”, we

outline the process to choose 7 crucial features out of 12 available options, and we provide explanations based on domain knowledge in physicochemical processes. All accessible properties (with links) that we can find at the amino acid level for preparation are listed in Table R4.

No.	Access	Feature name	Selected?
1	IPC	Isoelectric Point	√
2	Wikipedia	Polarity	√
3	Wikipedia and MolGpKa	Acidity and Alkalinity	√
4	RDKit	Hydrogen Bond Acceptor	√
5	RDKit	Hydrogen Bond Donor	√
6	RDKit	Octanol-Water Partition Coefficient	√
7	RDKit	Topological Polar Surface Area	√
8	Wikipedia	Relative Abundance	×
9	Wikipedia	Relative Molecular Mass	×
10	Website	Van Der Waals Volume	×
11	Wikipedia	Number of Rotatable Keys	×
12	RDKit	Number of Aromatic Rings	×

Table R4. Easily available features of amino acid level as optional inputs for our model.

Feature selection experiments:

Here, we humbly offer a succinct explanation of feature selection in this work. AI models can become more accurate and run faster by selecting the best subset of input information. Extremely fast feature selection is made possible by well-known model-independent methods such as maximizing correlation coefficient and maximizing mutual information. The model-independent feature selection methods, however, can only estimate the performance of AI models up to a certain point and might not be appropriate for all of them. Despite the potential time commitment, we employ model training-based methodologies to choose the best subset of features for our proposed model. Below, we describe the model-dependent feature selection approach.

Experiment process: To train and test our model, we remove a specific feature dimension from the dataset (note: not zero padding). We run 3 seeds for each feature dimension and determine the feature importance based on the average best-F1 score's negative value. All 12 optional features' importance values are obtained, and their z-scores are then computed.

Results: In Table R5, we display the mean of the importance for each kind of feature. We also show the z-scores and the final sort results. Following the sorting result, we gradually increase the feature

dimension from Topological Polar Surface Area (ranked 1st). The AUPR and F1 results peak once the feature of Octanol-Water Partition Coefficient (ranked 7th) is included. Thus, we ultimately settled on the seven physicochemical properties shown in the manuscript.

Removed feature	Best-F1 ↓	z-score ↑	Sort result	Selected?
Topological Polar Surface Area	84.25	1.56	1	✓
Isoelectric Point	84.44	1.25	2	✓
Hydrogen Bond Donor	84.57	1.04	3	✓
Polarity	84.83	0.62	4	✓
Hydrogen Bond Acceptor	85.00	0.35	5	✓
Acidity and Alkalinity	85.12	0.16	6	✓
Octanol-Water Partition Coefficient	85.40	-0.30	7	✓
Van Der Waals Volume	85.52	-0.49	8	×
Number of Aromatic Rings	85.57	-0.57	9	×
Number of Rotatable Keys	85.69	-0.76	10	×
Relative Abundance	86.08	-1.40	11	×
Relative Molecular Mass	86.13	-1.48	12	×

Table R5. Average F1 scores of our model after dropping each feature. We then calculate the importance z-scores for ranking. ↓ means lower Best-F1 score corresponds to more important feature. Instead, ↑ means higher z-score corresponds to more important feature.

Explanations:

To address your question about “why they choose the seven physicochemical properties”, we also justify some of selected features with domain knowledge in biology and chemistry mined from published papers.

Isoelectric Point (based on ref [R1]): The isoelectric point (IP) is the pH at which an amino acid carries no net electrical charge. Therefore, IP directly determines whether localized regions in proteins are positively or negatively charged in a solution environment at a specific pH. The interface of PPI (especially binding-type PPI) usually follows the principle of anisotropic attraction of charges, *i.e.*, PPI occurs only in the protein local areas where the overall charged situations are opposite.

Polarity (based on ref [R2], [R3]): The polarity of amino acid is defined by the hydrophilicity R groups. That is polar amino acids have hydrophilic R groups, while non-polar amino acids have hydrophobic R groups. The position of amino acids and their polarity have a great influence on the structure and function of the protein, especially determine the affinity of the two binding regions in the PPI process. Precisely, complementary alliance of hydrogen bonds of two proteins

Hydrogen Bond Acceptor/Donor (based on ref [R4], [R5]): A hydrogen bond is an intermolecular force formed by the interaction of a hydrogen atom that is covalently bonded to an electronegative atom (donor) with another electronegative atom (acceptor). Hydrogen bonds confers proteins specificity in protein-protein interactions, which greatly affects the binding free energy that determines the stability of protein complexes.

Octanol-Water Partition Coefficient (KOW) (based on ref [R6]): KOW is defined as the partition coefficient for the two-phase system consisting of n-octanol and water. Simply put, KOW serves as a measure of the relationship between lipophilicity and hydrophilicity. Ref [R6] reveals the role of hydrophobicity as a determinant of PPI. Moreover, surface hydrophobicity can be used to identify PPI interfaces.

Topological Polar Surface Area (TPSA) (based on ref [R7], [R8], [R9]): TPSA is defined as the surface sum over all polar atoms. Biological wet experiments and kinetic-based computer simulations consistently show a strong correlation between TPSA and the free energy of protein complexes.

[R1] Xia, X. Protein isoelectric point. *Bioinformatics and the Cell: Modern Computational Approaches in Genomics, Proteomics and Transcriptomics*, 207-219 (2007).

[R2] Bteich, M. An overview of albumin and alpha-1-acid glycoprotein main characteristics: highlighting the roles of amino acids in binding kinetics and molecular interactions. *Heliyon* **5**(11) (2019).

[R3] Cavalli, A. et al. Amino acids of the α 1B-adrenergic receptor involved in agonist binding: differences in docking catecholamines to receptor subtypes. *FEBS letters* **399**, 9-13 (1996).

[R4] Hubbard, R. E. & Haider, M. K. Hydrogen bonds in proteins: role and strength. *eLS* (2010).

[R5] Jones, S. & Thornton, J. M. Principles of protein-protein interactions. *Proceedings of the National Academy of Sciences* **93**, 13-20 (1996).

[R6] Young, L., Jernigan, R. & Covell, D. A role for surface hydrophobicity in protein-protein recognition. *Protein Science* **3**, 717-729 (1994).

[R7] Fernandes, J. & Gattass, CR. Topological polar surface area defines substrate transport by multidrug resistance associated protein 1 (MRP1/ABCC1). *Journal of medicinal chemistry* **52**, 1214-1218 (2009).

[R8] Elcock, A. H., Sept, D. & McCammon, J. A. Computer simulation of protein– protein interactions. *The Journal of Physical Chemistry B* **105**, 1504-1518 (2001).

[R9] Horton, N. & Lewis, M. Calculation of the free energy of association for protein complexes. *Protein Science* **1**, 169-181 (1992).

Supplementary notes:

We have revised the manuscript according to this comment.

1. We have clarified the different motivations for feature selection and feature importance

computation in the fifth paragraph of Section 2.3 (Line 256-261).

“For node features in protein graphs, we first select seven important features from twelve residue-level feature options (see Supplementary Table 4) that are easily available. The feature selection process (see Supplementary File 1 for details) produces the optimal set consisting of seven features to ensure that our model peaks at both AUPR and best-F1 scores. Here, we list the selected seven residue-level physicochemical properties in Fig. 3d and discuss their importance for different types of PPIs to both better interpret our model and discover enlightening biomarkers for PPI interface.”

2. We have added a description of the feature selection process (7 important features out of 12 optional ones) and the selection results in the Supplementary File 1.

(3) In the discussions, the authors mentioned that a future work may be considered at atomic level. I think there is a middle-scale view, which is at domain-scale. Maybe this can be discussed. Surely, if the authors feel this is unnecessary or not proper in this paper, they may choose to ignore this.

We would appreciate your constructive comments on the future work of hierarchical graph learning. We strongly agree with you that the protein domain may offer a useful middle-scale view on the PPI issue. Actually, from both structural and functional standpoints, protein domains fall somewhere in the middle of amino acids and proteins. Protein domains are compact, foldable, three-dimensional structures made of amino acid residues. The three-dimensional structure of the complete protein is made up of several protein domains acting as structural building blocks. When it comes to function, each domain oversees expressing a certain protein function. The specificity of protein activities is produced by the assembly of various domains, which also controls the existence or absence of PPIs and hot spots at the PPI interface. **Therefore, both in terms of structures and functions, the protein domain represents a crucial middle scale for the PPI problem.**

Our authors believe that it would be more appropriate to introduce the domain scale in a separate work. To the best of our knowledge, most domain annotations will be acquired through computational prediction tools like InterPro [R10] and SMART [R11], which might offer inaccurate domain data. Although it makes intuitive sense that the protein domain is an intermediary take up in the PPI problem, the authors had not previously given the idea much thought (at least not until the reviewer brought it up), mostly due to the dearth of trustworthy domain annotations. There must be a lot more work to be done on top of the HIGH-PPI to figure out how to create an end-to-end AI model for the residue-domain-protein hierarchy and guarantee the consistency of representation between adjacent views. In addition, the introduction of the middle scale view can be considered as the process of data upscaling, which could result in more valuable information or on the contrary, the loss of accuracy due to data redundancy. Based on the findings of the HIGH-PPI evaluation alone, our downscaled double-viewed model, which temporarily ignores the domain-scale, performs satisfactorily and we anticipate that the addition of new data will maintain this pervasiveness. **Nevertheless, we respectfully recommend further research on multi-view hierarchical learning for PPI prediction.**

About future work: ① (Information benefits) It has been demonstrated that the proposed double-viewed hierarchical model, which is based on the high degree of information complementarity between the data of two views, benefits from both views. For instance, the PPI network (top view)

will suggest active protein information (*i.e.*, with a high network degree from top view) to the bottom view. Similarly, the bottom view helps PPI network to complement residue-level fragmentation knowledge, *i.e.*, the knowledge of the functions performed by residue fragments for a particular PPI instance. The two current perspectives will benefit from the accuracy of the domain annotations because it provides data on functional residue segments that are crucial to PPIs. However, when the domain scale is employed as a separate view, data unreliability could spread to other views and may impair the hierarchical model as a whole. Therefore, for the information gain in the hierarchical model, an early reliability assessment of the available domain annotations is required. ② (Supervised information at the residue level) HIGH-PPI now enables interpretable identification for PPI-related functional sites without supervising domain data. Therefore, directly supervising the selection of significant functional sites is a straightforward way for the double-viewed hierarchical model to gain from domain information. Precisely, a well-designed regularization is required to guarantee that all functional sites, discovered by HIGH-PPI, belong in the prepared domain database. The domain regularization and the PPI prediction loss form a flexible trade-off of learning objectives, which can appropriately tolerate the domain annotation unreliability. ③ (Computational efficiency) It is worth mentioning that the explosion of data required for multi-view hierarchical modeling necessitates the use of lightweight backbones for quick training and prediction. Our suggested hierarchical graph learning backbone, a memory and compute efficient backbone for simultaneously learning the structure-function relationship, may therefore provide insight for future studies. ④ (Protein structure modeling) The research on protein structure modeling, which is critical for understandable mining of essential interaction areas, needs to be continued in the upcoming work.

[R10] Hunter, S. et al. InterPro: the integrative protein signature database. *Nucleic acids research* **37**, D211-D215 (2009).

[R11] Letunic, I. & Bork, P. 20 years of the SMART protein domain annotation resource. *Nucleic acids research* **46**, D493-D496 (2018).

Supplementary notes:

1. We have added an overview discussion of the domain scale in the last paragraph of Section 3.2 (Line 377-388).

“(2) Protein domain information may be beneficial for hierarchical models. We clarify the core ideas here and provide detailed description in Supplementary File 2. Domains are distinct functional or structural units in proteins and are responsible for PPIs and specific protein functions. Both in terms of structures and functions, the protein domain can represent a crucial middle scale for the PPI hierarchy. However, to our knowledge, true (native) domain annotations are not easily available and predicted ones are usually retrieved from computational tools, which inevitably leads to the data unreliability. If we employ the domain scale as a separate view, data unreliability may spread to other views and impair the entire hierarchical model. On this basis, we prefer to recommend domain annotations as supervised information at the residue level. Precisely, a well-designed regularization is required to guarantee that all

functional sites, discovered by HIGH-PPI, belong in the prepared domain database. The domain regularization and the PPI prediction loss form a flexible trade-off of learning objectives, which can appropriately tolerate the domain annotation unreliability.”

2. We have also added detailed discussion of the domain scale in the Supplementary File 2.

(4) Although the authors have shared the data and code, I still have no time to validate/reproduce the results myself. However, I managed to browse the data and have a suggestion. The authors should share the data in pure text format, like csv/tsv/txt etc., but not the numpy/npz format, which is not convenient for quick browsing.

Thanks for your kind comment. Clear data visualization for convenient browsing is also one of our goals. To address this issue, we have converted some of the files into the csv or txt format which the audience may wish to view quickly. The audience can access the data file here for quick browsing. We have presented some of the screenshots of those files below (Table 1R-3R). To facilitate viewing the graph-structured data of proteins, we generated adjacency matrix heatmaps for some proteins in the SHS27k dataset (see examples in Table 4R).

ReadMe: This sheet shows the ground-truth labels and predicted results of PPIs for the test set. We present the Ensembl ids of the protein pairs and the presence (1) or absence (0) of 7 type PPIs.

Index	Protein_1	Protein_2	Ground-truth labels							Predicted results						
			Reaction	Binding	Phnol	Activation	Inhibition	Catalysis	Expression	Reaction	Binding	Phnol	Activation	Inhibition	Catalysis	Expression
0	9606.ENSP0000027567	9606.ENSP00000256720	0	0	1	0	0	0	0	0	0	1	0	1	0	0
1	9606.ENSP0000005340	9606.ENSP00000210113	0	0	0	1	0	1	0	1	0	1	0	1	0	1
2	9606.ENSP0000005257	9606.ENSP00000229264	1	1	0	1	0	1	0	1	1	0	1	0	1	0
3	9606.ENSP00000174618	9606.ENSP00000261254	0	0	0	1	0	1	0	1	0	0	1	0	1	0
4	9606.ENSP0000006104	9606.ENSP00000258743	0	1	0	0	0	0	0	0	1	0	0	0	0	0
5	9606.ENSP00000172229	9606.ENSP00000256010	1	1	0	1	0	1	0	1	1	0	1	0	1	0
6	9606.ENSP00000220751	9606.ENSP00000227758	0	0	0	1	1	1	0	0	0	1	1	1	0	0
7	9606.ENSP00000215071	9606.ENSP00000249060	1	1	0	0	0	0	1	0	1	1	0	0	0	1
8	9606.ENSP00000180173	9606.ENSP00000221086	1	1	0	1	0	1	0	1	1	0	1	0	1	0
9	9606.ENSP00000215812	9606.ENSP00000232461	1	1	0	1	0	1	0	1	0	1	0	1	0	0
10	9606.ENSP00000199551	9606.ENSP00000293469	0	0	1	0	0	0	0	0	0	0	1	0	0	0
11	9606.ENSP00000221930	9606.ENSP00000256509	1	1	0	1	0	1	0	1	0	0	1	0	1	0
12	9606.ENSP00000078429	9606.ENSP00000229955	1	1	0	0	0	1	0	1	1	0	0	0	1	0
13	9606.ENSP00000204679	9606.ENSP00000247461	0	0	0	1	0	0	0	0	1	0	1	0	0	0
14	9606.ENSP00000250008	9606.ENSP00000262187	1	0	0	0	0	0	1	0	1	0	0	0	0	1
15	9606.ENSP00000013125	9606.ENSP00000247608	0	0	0	1	0	0	0	0	0	0	1	0	0	0
16	9606.ENSP00000178640	9606.ENSP00000246792	0	0	0	0	0	0	0	1	0	0	0	0	0	1
17	9606.ENSP00000220751	9606.ENSP00000250559	0	0	0	1	0	1	0	0	0	0	1	0	1	0
18	9606.ENSP00000160382	9606.ENSP00000216181	0	0	0	1	0	1	1	0	0	0	1	0	0	1
19	9606.ENSP0000025792	9606.ENSP00000256797	1	1	0	1	0	1	0	1	1	0	0	0	1	0
20	9606.ENSP00000244741	9606.ENSP00000281709	0	0	1	0	0	0	0	0	0	1	0	0	0	0
21	9606.ENSP00000215971	9606.ENSP00000229758	0	1	1	0	0	0	0	0	1	0	0	0	0	0
22	9606.ENSP00000075403	9606.ENSP00000246841	0	1	0	0	0	0	0	0	1	0	0	0	0	0
23	9606.ENSP00000167586	9606.ENSP00000252252	0	0	0	1	0	0	0	0	0	0	1	0	0	0
24	9606.ENSP00000000412	9606.ENSP00000244458	0	0	0	0	0	0	1	0	0	0	0	0	0	1
25	9606.ENSP00000215781	9606.ENSP00000262345	0	1	0	0	0	0	0	0	1	0	0	0	0	0
26	9606.ENSP00000011610	9606.ENSP00000338703	1	1	0	0	0	0	1	0	1	0	0	0	1	0

Figure R1. The screenshot of the csv file presenting predicted results.

Amino acid	Feature type								
	IP	P	A&A	HBA	HBD	KOW	TPSA		
ALA	6.00	0.00	0.00	3.00	3.00	-0.67	63.32	IP: Isoelectric Point	
CYS	5.02	1.00	0.00	3.00	3.00	-0.67	63.32	P: Polarity	
ASP	2.77	2.00	1.00	3.00	3.00	-1.13	100.62	A&A: Acidity & Alkalinity	
GLU	3.22	2.00	1.00	3.00	3.00	-0.74	100.62	HBA: Hydrogen Bond Acceptor	
PHE	5.48	0.00	0.00	2.00	2.00	0.64	63.32	HBD: Hydrogen Bond Donor	
GLY	5.97	1.00	0.00	2.00	2.00	-0.97	63.32	KOW: Octanol-Water Partition Coefficient	
HIS	7.47	3.00	2.00	3.00	3.00	-0.64	92.00	TPSA: Topological Polar Surface Area	
ILE	5.94	0.00	0.00	2.00	2.00	0.44	63.32		
LYS	9.59	3.00	2.00	3.00	3.00	-0.47	89.34		
LEU	5.98	0.00	0.00	2.00	2.00	0.44	63.32		
MET	5.74	0.00	0.00	3.00	2.00	0.15	63.32		
ASN	5.41	1.00	0.00	3.00	3.00	-1.73	106.41		
PRO	6.30	0.00	0.00	2.00	2.00	-0.18	49.33		
GLN	5.65	1.00	0.00	3.00	3.00	-1.34	106.41		
ARG	11.15	3.00	2.00	3.00	5.00	-1.34	125.22		
SER	5.68	1.00	0.00	3.00	3.00	-1.61	83.55		
THR	5.64	1.00	0.00	3.00	3.00	-1.22	83.55		
VAL	5.96	0.00	0.00	2.00	2.00	0.05	63.32		
TRP	5.89	0.00	0.00	2.00	3.00	1.12	79.11		
TYR	5.66	1.00	0.00	3.00	3.00	0.35	83.55		

Figure R2. The screenshot of the csv file presenting input amino acid level features.

Protein ID	Sequence
9606.ENSP00000000253	MGLTVSALFSRIFGKQMRILMVLGLDAAGKTTILYKLLQGEIVTPTIGFNVEYVKNICFTVVDVGGQKIRPLWRHYFQNTQGLIFVDSNDRERVQESADELQKMLQEDELDRDA
9606.ENSP000000250971	MALWMRLLPLLLALLWGPDDAAAFVNHLCGSHLVEALYLVCGERGFFYTPKTRREAEDLQVGGQVLEGGGPGAAGSLQPLALEGSLQKRGIVEQCTTICSLYQLENYCN
9606.ENSP00000019317	MTECFLLPTSSPEHRRVVEHSGSLTRTPSSEISPTKFPQLYRTGEPSPHDILHEPPDVSDDEKDHOKKGGKFKKKEKRTGYAAFQEDSSGDEAESPSKMRKRGHVFKKPSPSKK
9606.ENSP000000126373	MQQAPQPVYFTSEENSPKWRGLLVSALREYQEQVHPILSANEESLYVIEELFQLLNKLCMAQPRTVQDVEERVQKTFPHPIDKWAJADAQSAIEKRRKRNPLLLPVDKGGHPSLKEVLGY
9606.ENSP00000012443	MAAMAEGERTECAEPPRIEPPADGALKRAEELKTQANDYFKAKDYENAIKFSYQAEINPSNAIYGNRSLAYLRTECYGYALGDATRAIELDKKYLKGYRRAASNMAJGKFRALRE
9606.ENSP000000005340	MAGSSTGGGVGTEKIVTILDEEETPYLVKIPVPAERITLGDFKSVLQRPAGAKYFKKMDQDFVYKKEISDDNARLPCFNGRVSWLVSSDNQPPEMAPPVHEPRAELAPPAPLPP
9606.ENSP000000211287	MSLRKKGFYKQDVNKTAWELPKTYVSPTHVGSAGYGSVAIDKRSRGEKVAIKLSRPFQSEIFAKRAYRELIILKHMQHENVIGLLDVFTPASSLRNFYDFYLVMPFQITLQKIMG
9606.ENSP000000178640	MLWLAGPPFAMENQVLRIRKIPNSGAVDWTVHSGPQLFRIDVLDVIGQVLEATTAFVEYEDDGRITVRSDEEMKAMLSYYSYSTMVEQVNGQLIEPLQIFRACKPPGFRNHHI
9606.ENSP00000020673	MAAQAMHPCSEODCAISPPICPRRVLPEGPVPSPPASMYGTSGLRRVAGPGRGRELGRVAPCTPLRGPSPSRVAPSPWAPSSPTGQPPPPQAQSSVYFRFVKEAKVPLNGLPAIN
9606.ENSP000000978429	MTLFSMACCLADFKESKRINAEIEQLRRDKRDARRELKLLGTGESGKSTFIKQRIHAGYSEEDKRGFTKLVYQNFITAMQAMIRAMELTKLYKYEQNKANALIRFVDVFR
9606.ENSP000000202677	MFSRHSHIGDVKKSTQKVLDPKRDVLRILKILRALLDNVDANDLQKFFETNYVQYTFYFENFIALENSLKLGKNNKSGRELDHILFEKILQFLPERIFRWHYQMGSTLKKLHITGN
9606.ENSP000000215659	MNSPPPARSGFYRQEVTKTAWEVRAVYRDLQPVGSGAYGAVCSAVDGRITGAKVAIKLIRPFPQSELFAKRAYRELIILKHMRIHENVIGLLDVFTDEFLDDFTDFYLVMPFMGTDLG
9606.ENSP000000219255	MAQRPTPARSPDSEIVKSKFDAEFRFALPRASVGFQFSPRLRAYHQPLDVLGYTDAHGDLPLTNDLHRALASGPPPLRLLVQKRAEADSSGLAFASNSLQJQKGLLRF
9606.ENSP000000219409	MLGLDACELGALQLELLRLALCARVLLADKEGGPPAVDEVLEDAVPEYRAGPKKSLERIQDLDDRSRLAKYKRVLLGPLPFAVDPSSLFNVQVTRLLTLESAQPGPVVMDLGDGLAVL
9606.ENSP000000230440	MNRAGLGEVYPPGNYGNYANSOYSACEEENRLETSLSKSYTAIKLSIEIGHEVKTQNKLLAEMDSQFDSITGFLGKTMGKILSRGQSTKLLCYMMLFSLVFFIHWIILR
9606.ENSP000000221148	MDDKAFKELDQWVEQLNECKQLNENQVRLCEKAKEILTKESNVQEVRCPVTVCGDGVHGQFHDLMELFRIGGKSPDNYLFGDGYVDRGYVSYVTVTLVALKRVYPERUILRON
9606.ENSP00000022345	MTTYRAIPSDGVDLAASCGARVQDVLPGPHTDYAPLQFVAQNSGMSQPLGESPATATATATATRPSPTTPAMPKMGVRAVADWPPIKREALREHNSPSPSQDIDGKATKMAHSM
9606.ENSP000000229794	MNQERPTFYRQELNKTIWEVPERYQNLSPVGSAGYGSVAADFTKLTGLRVAVKLSRPFQSIHAKRTYRELRLLKHMKHENVIGLLDVFTPARSLEEFNDVYVTLHMLAIDLNMAHVC
9606.ENSP000000228367	MDDLLALLADLESTTHSKRPVFLSEETPYVPTQNHVYQELAVPPVPPPPSSEALNGTLLDPLDQWQPSSRFHQQPQSSPVYGSAACTSSVSNPQDSEVGPSCRVGEEHHVSYFNMK
9606.ENSP000000223567	MDPLPQQTHKQVHEIQCMGRLEATDKQVHVIVENEQASIDQFSLRELEILSKPEPNKQNAIRLVQDLKYDVQHLQALRNFQHRHAREQERQREELLSKRTITNDSIDTTPIM

Figure R3. The screenshot of the csv file presenting protein sequences.

Figure R4. Examples for the adjacency matrix heatmaps from the inside-of-protein view. Each heatmap shows the way amino acids are linked with in a protein.

(5) There are several minor confusing writings in the text and math, I think the authors should verify them. I also suggest the authors to proof read the manuscript again, as some kind of "hurry" can be felt when reading its current version.

Thanks for pointing this out. We agree and have made the following edits.

On Page 6, legends of the fig1 "SEPRINA3" should be "SERPINA3"? Please notice the spelling

We have replaced the typo "SEPRINA3" with "SERPINA3" in the fourth Line of the Fig. 1 legend.

On Page 18, Line 332, I think it is "3.2", not "3.1"

We have changed the section number in Line 360 (originally Line 332) to "3.2".

On Page 20, Line 363, "gb \in gt", this does not make sense. I think it should be "gb \in Vt", according to the words on Line 362

We have changed the notation " $g_b \in g_t$ " to " $g_b \in V_t$ " in Line 411 (originally Line 362).

On Page 23, Line 433, " $G_1 \subseteq G_1$ ", this does not make sense, please verify.

We have changed the wrong notation " $G_1 \in G_1$ " to " $G_s \in G_1$ " in Line 484 (originally Line 433)

We apologize for the confusion caused by the typos above. As a precaution, we have performed a thorough double-check of the writing details in our manuscript.

Reviewer #2 (Remarks to the Author):

Comments to the Author:

The manuscript 'Hierarchical Graph Learning for Protein-Protein Interaction' submitted by Gao et al deals with a novel hierarchical graph learning based method allowing the authors to predict interactions for given protein pairs and key residues for their interactions. The authors used multi-type human protein-protein interactions (PPI) and showed that their method demonstrates high accuracy and robustness in predicting PPIs. At the core of their method is a synergistic predictive effect of two levels: the level of the structural representations of the protein itself (lower level) and the level of the PPI network (upper level). By mutually sharing the information learned in each of the lower and upper levels, it leads to better structural representations of proteins and learning relationships between proteins. Their method is novel in that it models the natural PPI relationship by hierarchically linking both levels. In addition, it is also interesting to graphically represent protein and network structures using Graph Neural Networks (GNN) for this modelling. Furthermore, the authors have rigorously evaluated the robustness of their method from various viewpoints, increasing the reliability of their method. The manuscript is well written and well-illustrated. I think that it is worthy of being published in Nature Communication. I only have some minor questions and comments:

Thanks for your in-depth explanation of the core methodology of our work and for your recognition of our contribution. To address your constructive comments, we provide point-by-point responses below.

Minor questions and comments:

- Page 7, in the section 2.1, line 134-136: The proposed method is trained and evaluated using a dataset consisting of 1,690 proteins and 7,624 PPIs. By simple calculation, one protein is involved in 4.5 PPIs ($7,624 \text{ PPIs} / 1,690 \text{ proteins} = 4.511\dots$). According to Park and Marcotte (Nat Methods. 2012 December; 9(12): 1134–1136. doi:10.1038/nmeth.2259.), PPI prediction methods tend to perform much better on test pairs that share the same proteins with the training dataset than those that do not. Although the generalization evaluation (OOD) has been done, it would be useful for readers to show the overall performance and robustness evaluation in C2 and C3 among 3 distinct classes proposed by Park and Marcotte.

Thank you for pointing out the interesting and enlightening way for revisiting the model performance. In response to your suggestions, we have designed experiments for the generalization evaluation (**GE**), which assesses the model's capacity to **handle OOD data**. Furthermore, we performed robustness evaluation (**RE**) by manually adding **perturbations to the OOD dataset**.

Experimental protocol:

There’re following tips to clarified before the experiments:

First, we created 3 data partitions based on the SHS27k dataset. Among the test sets split by 3 data partitions, each of them shares a range of 3 different test pairs including C_1 , C_2 , and C_3 . Here, C_1 stands for the percentage of PPIs of which **both proteins were present** in a training set (Class 1), C_2 stands for the percentage of PPIs of which **one of (but not both) proteins was present** in the training set (Class 2), C_3 stands for the percentage of PPIs of which **neither protein was present** in the training set (Class 3).

Second, we didn’t directly employ the random or OOD (BFS, DFS) partitioning methods presented in the manuscript to create the three partitions because of two limitations: 1) the C_3 from random partitioning is almost 0%, and 2) the C_1 from OOD partition is almost 0%, thus neither the random nor OOD alone is suitable for statistical evaluation of **3 distinct classes**. To overcome this issue, we combine the random and OOD partitioning methods to ensure that the size of C_1 , C_2 , and C_3 is considerably balanced.

Third, the 3 datasets were directly used in the experiments of GE. In RE, we additionally added perturbations to the training sets of the created 3 datasets and remained the test sets unchanged.

Generalization evaluation experiments: We train and test the model on the 3 produced datasets. For each dataset, we show model performance on the overall dataset and on 3 distinct classes. For each experiment, we run 5 seeds and report the average F1 score.

Robustness evaluation experiments: We follow the experimental protocol for robustness evaluation in the manuscript. As for the 3 produced datasets, we add random perturbations (perturbation ratio is 0.2) on their training sets and retain their test sets unmodified. For each dataset, the model’s performance is displayed in overall fashion and respective fashion on 3 distinct classes. For each experiment, we run 5 seeds, and then report the average F1 score.

Results and discussions:

Dataset	$C_1/C_2/C_3$ (%)	Overall		Class 1		Class 2		Class 3	
		Ours	Bl	Ours	Bl	Ours	Bl	Ours	Bl
1	27/51/22	75.05	70.77	87.22	84.12	78.31	73.20	52.56	48.77
2	36/48/16	77.40	72.17	87.46	84.53	77.40	71.19	54.79	47.28
3	21/64/15	75.94	71.48	87.29	84.43	77.09	72.94	55.17	47.09
Avg. gain		4.40		2.96		5.16		6.46	

Table R6. Generalization experiments. We show the averaged F1 scores of our model and a strong baseline (GNN-PPI) across 5 seeds. ‘Bl’ and ‘Ours’ represent the solid baseline method

and HIGH-PPI, respectively. Avg. gain shows the performance gain between our method and the baseline method.

Dataset	$C_1/C_2/C_3$ (%)	Overall		Class 1		Class 2		Class 3	
		Ours	BI	Ours	BI	Ours	BI	Ours	BI
1	27/51/22	63.42	56.21	81.13	73.91	62.75	55.89	43.26	35.26
2	36/48/16	66.23	59.72	80.52	73.63	61.17	56.59	45.40	37.79
3	21/64/15	63.75	56.57	80.44	73.05	62.53	55.70	45.62	37.24
Avg. gain		6.97		7.16		6.09		8.00	

Table R7. Robustness experiments. We show the averaged F1 scores of our model and a strong baseline (GNN-PPI) across 5 seeds. ‘BI’ and ‘Ours’ represent the solid baseline method and HIGH-PPI, respectively. Avg. gain shows the performance gain between our method and the baseline method.

Figure R5. Bar chart visualization for the GE and RE experiments.

We present the following **core and secondary findings** for above generalization evaluation (GE) and robustness evaluation (RE).

➤ **Core findings:**

We come to the same conclusion as Park and Marcotte did [R12]. In both RE and GE, there is a noticeable difference in model test performance across the 3 distinct classes of test pairs. Particularly, on Class 1 test pairs, both models performed best, on Class 2 test pairs they were second best, and on Class 3 test pairs they were poorest. Furthermore, we found that for each model, the class proportion (*i.e.*, $C_1/C_2/C_3$) had an impact on the overall performance of the model despite having little effect on performance on the respective classes. Thus, it seems that the proportion of the three test pair classes as well as the percentage of unknown proteins in the test sets may both have a significant role in determining the degree of OOD in the dataset. We appreciate your valuable review, which will allow us to better define the **OOD problem in pairwise prediction challenges**.

➤ **Secondary findings:**

- (1) In GE, our method performs better on all occasions we presented. As can be seen on the Avg. gain, as the prediction difficulty increases (Class: 1→2→3), the superiority of our method tends to be more significant (Avg. gain: 2.96→5.16→6.46), demonstrating that our model could better handle OOD protein pair data.
- (2) RE is essentially a robust test of the model using OOD data, in which the models performed significantly worse than they do in GE. Even for the easy-to-predict (Class 1) test pairs, RE leads to a 10.83 drop in the average performance score compared to GE.
- (3) Our model's performance in GE has a significant advantage than that in RE. The advantage is reflected in the overall (Avg. gain: 4.40 vs. 6.97) and 3 distinct classes (Avg. gain: (2.96, 5.16, 6.46) vs (7.16, 6.09, 8.00)). This suggests that our model can better handle the most realistic scenarios of PPI problem, where both OOD and perturbations are present.

[R12] Park, Y. & Marcotte, E. M. Flaws in evaluation schemes for pair-input computational predictions. *Nature methods* **9(12)**, 1134-1136 (2012).

Supplementary notes:

We have revised the manuscript according to this comment.

1. We have added the recommended reference as [61] in the manuscript and added the description of mentioned generalization experiments in the fourth paragraph of Section 2.2 (Line 194-206).

“Furthermore, we follow Park and Marcotte⁶¹ to explore the differences of model performance on 3 kinds of PPI pairs with different degrees of OOD. Specifically, C_1 stands for the percentage of PPIs of which both proteins were present in a training set (Class 1), C_2 stands for the percentage of PPIs of which one of (but not both) proteins was present in the training set (Class 2), C_3 stands for the percentage of PPIs of which neither protein was present in the training set (Class 3). The detailed experimental protocol has been presented in the Supplementary File 3. We come to the same conclusion as Park and Marcotte did⁶¹. There is a noticeable difference in model test performance across the 3 distinct classes of test pairs. Particularly, on Class 1 test pairs, both models (HIGH-PPI and GNN-PPI) perform the best, on Class 2 test pairs they are the second best, and on Class 3 test pairs they are the poorest. Furthermore, we find that for each model, the class proportion (*i.e.*, $C_1/C_2/C_3$) had an impact on the overall performance of the model despite

having little effect on performance on the respective classes. Thus, it seems that the proportion of the three test pair classes (supplementary Table 6) as well as the percentage of unknown proteins (Fig. 2c) in the test sets may both have a significant role in determining the degree of OOD in the dataset.”

2. We have added the detailed experimental protocol and results in the Supplementary File 3.

- Page 20, in the section 4.1, line 370-372: The presence or the absence of contact between a pair of residues is determined by their C_α-C_α physical distance, and the threshold is defined as ~10 Å. Please explain how you defined this threshold value. Is this value (~10 Å) a reasonable distance?

Thanks for your constructive comment. We concur that the cutoff distance (threshold) is worth of discussion. Our explanation in this case is based on understanding of biophysics and machine learning. For ease of use, we abbreviate the cutoff distance as c_d .

In terms of deep graph learning, the major goal of protein representation with graph structure data is to define protein features linked to 3D structure in a memory and computation efficient manner. The **inter-residue forces** are necessary for **protein structural stability**, which in turn defines the **protein properties**, should normally be maintained by the selected c_d [R13, R14]. The ideal c_d , however, is inconclusive and depends on variables including a wet experimental environment and protein family differences, according to the majority of prior research [R15]. Pioneer work selects $c_d = 8 \text{ \AA}$ (*i.e.*, residues only within 8 Å can be connected) as the ideal range for **residue-residue interactions (RRIs)**. This range can be used to characterize a variety of protein properties, including their hydrophobic behavior [R16, R17], folding mode [R18, R19, R20], stability upon mutations [R21], and thermal stability [R22]. Nevertheless, some studies have found that closer distances [R23] or longer distances [R24] are appropriate. In contrary, a bigger c_d (more than 8) reduces the neglect of RRIs, allowing the graph data to capture more protein features. For instance, $c_d = 8 \text{ \AA}$ is adequate to reflect protein hydrophobic behaviors but falls short of capturing other traits including protein geometric properties [R24]. But c_d cannot be endlessly big since joining weakly interacting residues results in erroneous predictions of the features of proteins.

Based on the literature research and logical consideration, one hypothesis is that the optimal c_d can vary in different models and as well as in relevant datasets, thus c_d needs to be identified case by case. Therefore, we chose the $c_d = 10 \text{ \AA}$ based on an empirical study (Figure R6). In order to choose a suitable c_d , we generated 11 sets of protein adjacency with a parameter sweep of c_d between 5.0 and 15.0 at 1.0 intervals. We ran 3 seeds for each c_d and calculate the average results as F1 scores. This empirical study reveals that an optimal cutoff distance (c_d) is around ~10 Å in our model, also the model’s performance grows quickly at $c_d = \sim 7 \text{ \AA}$ and declines at $c_d = \sim 13 \text{ \AA}$. Even though we selected $c_d = 10 \text{ \AA}$, it is important to note the incredibly narrow range of F1 values. The optimal cutoff distance (c_d) can be selected from an aggregation between 9-12 Å because the overall F1 scores vary from 0.8621 to 0.8633 for the c_d between 9-12 Å, which also agrees with the literature observations in the paragraph above.

Figure R6. Experimental results for optimal cutoff distance (c_d) selection.

[R13] Srivastava, D., Bagler, G. & Kumar, V. Graph Signal Processing on protein residue networks helps in studying its biophysical properties. *bioRxiv* (2021).

[R14] Gromiha, M. M. & Selvaraj, S. Inter-residue interactions in protein folding and stability. *Progress in biophysics and molecular biology*, **86(2)**, 235-277 (2004).

[R15] Yan, W. et al. The construction of an amino acid network for understanding protein structure and function. *Amino acids*, **46(6)**, 1419-1439 (2014).

[R16] Manavalan, P. & Ponnuswamy, P. K. Hydrophobic character of amino acid residues in globular proteins. *Nature*, **275(5681)**, 673-674 (1978).

[R17] Ponnuswamy, P. K. Hydrophobic characteristics of folded proteins. *Progress in biophysics and molecular biology*, **59(1)**, 57-103 (1993).

[R18] Debe, D. A. & Goddard, W. A. First principles prediction of protein folding rates. *Journal of Molecular Biology*, **294(3)**, 619-625 (1999).

[R19] Gromiha, M. M & Selvaraj, S. Comparison between long-range interactions and contact order in determining the folding rate of two-state proteins: application of long-range order to folding rate prediction. *Journal of molecular biology*, **310(1)**, 27-32 (2001).

[R20] Kocher, J. P. A., Rooman, M. J. & Wodak, S. J. Factors influencing the ability of knowledge-based potentials to identify native sequence-structure matches. *Journal of molecular biology*, **235(5)**, 1598-1613 (1994).

[R21] Gromiha, M. M. et al. Role of structural and sequence information in the prediction of protein stability changes: comparison between buried and partially buried mutations. *Protein Engineering* **12(7)**, 549-555 (1999).

[R22] Gromiha, M. M. Important inter-residue contacts for enhancing the thermal stability of thermophilic proteins. *Biophysical Chemistry*, **91(1)**, 71-77 (2001).

[R23] Yang, L., Song, G. & Jernigan, R. L. Protein elastic network models and the ranges of cooperativity. *Proceedings of the National Academy of Sciences*, **106(30)**, 12347-12352 (2009).

[R24] Duarte, J. M. et al. Optimal contact definition for reconstruction of contact maps. *BMC bioinformatics* **11(1)**, 1-10 (2010).

Supplementary notes:

We have revised the manuscript according to this comment.

1. We have added the description of the optimal cutoff distance experiment in the third paragraph of Section 4.2 (Line 420-422).

“We perform a sensitivity analysis (see Supplementary Fig. 8) and find that our model produces similar results when trained on contact maps with cutoff distances ranging between 9 Å to 12 Å. Finally, we choose the optimal cutoff distance of 10 Å, which allows our model to peak its performance.”

2. We have added experimental results for selecting the optimal cutoff distance in the figure form (Supplementary Figure 8) to the Supplementary Information file.

- Page 10, in the section 2.3, line 177: “BFS-0.3 denotes that the test set involving 30% known proteins ...” should be “BFS-0.3 denotes that the test set involves 30% known proteins ...”.

Thanks for pointing this out. We agree and have corrected the word “involving” to “involves” in Line 188 (originally Line 177).

- Page 12, in the section 2.3, line 199: “GNNPPI” should be “GNN-PPI”.

Thanks for pointing this out. We agree and have changed the wrong notation “GNNPPI” to “GNN-PPI” in Line 223 (originally Line 199).

- Page 20, in the section 4.1, line 366: Is “(i.e., A_b, X_b, A_t, A_t)” a mistake of “(i.e., A_b, X_b, A_t, X_t)”?

Thanks for pointing this out. We agree and have changed the wrong notation “ A_b, X_b, A_t, A_t ” to “ A_b, X_b, A_t, X_t ” in Line 414 (originally Line 366).

- Page 23, in the section 4.5, line 433: Is “ $G_1 \subseteq G_1$ ” a mistake of “ $G_s \subseteq G_1$ ”?

Thanks for pointing this out. We agree and have changed the wrong notation “ $G_1 \in G_1$ ” to “ $G_s \in G_1$ ” in Line 484 (originally Line 433).

We thank you for your careful review and apologize for the confusion caused by the typos above. As a precaution, we have performed a thorough double-check of the writing details in our manuscript.

Reviewer #1 (Remarks to the Author):

I think the authors have addressed all my concerns. This can be accepted now.

Reviewer #2 (Remarks to the Author):

The authors have sincerely addressed my concerns in a clear and satisfactory manner. There is no need for me to review their revised manuscript again. Therefore, unless the other reviewer finds any concerns in the revised version, I recommend publishing their excellent work.